

# Organic carbon at a remote site of the western Mediterranean Basin: composition, sources and chemistry during the ChArMEx SOP2 field experiment

Vincent Michoud[1,2,3], Jean Sciare[4,5], , Stéphane Sauvage[1,2], Sébastien Dusanter[1,2,6], Thierry
Léonardis[1,2], Valérie Gros[4], Cerise Kalogridis[4], Nora Zannoni[4], Anaïs Féron[4], Jean-Eudes
Petit[4,7,*], Vincent Crenn[4], Dominique Baisnée[4], Roland Sarda-Estève[4], Nicolas Bonnaire[4],
Nicolas Marchand[8], H. Langley DeWitt[8], Jorge Pey[8,**], Aurélie Colomb[9], François Gheusi[10],
Sonke Szidat[11], Iasonas Stavroulas[5], Agnès Borbon[3,***], Nadine Locoge[1,2]
[1] Mines Douai, SAGE, F-59508, Douai, France
[2] Université de Lille, 59655, Villeneuve d'Ascq, France
[3] LISA, UMR-CNRS 7583, Université Paris Est Créteil (UPEC), Université Paris Diderot (UPD), Institut
Pierre Simon Laplace (IPSL), Créteil, France
[4] LSCE, IPSL, CEA et Université de Versailles, CNRS, Saint-Quentin, France
[5] The Cyprus Institute, Energy Environment Water Research Center, Nicosia, Cyprus
[6] School of Public and Environmental Affairs, Indiana University, Bloomington, IN, USA
[7] INERIS, 60550 Verneuil-en-Halatte, France
[8] Aix Marseille Univ, CNRS, LCE, Marseille, France
[9] LaMP, UMR-CNRS 6016, Clermont Université, Université Blaise Pascal, Aubière, France
[10] Laboratoire d'Aérologie, Université de Toulouse, CNRS, Toulouse, France
[11] Department of Chemistry and Biochemistry & Oeschger Centre for Climate Change Research,
University of Bern, Bern, Switzerland
*now at Air Lorraine, 20 rue Pierre Simon de Laplace, 57070 Metz, France
**now at the Geological Survey of Spain, 50006 Zaragoza (Spain)
***now at LaMP, UMR-CNRS 6016, Clermont Université, Université Blaise Pascal, Aubière, France
Corresponding authors :
V. Michoud (vincent.michoud@lisa.u-pec.fr)
S. Sauvage (stephane.sauvage@mines-douai.fr)
**Abstract**



The ChArMEx (Chemistry and Aerosol Mediterranean Experiment) SOP2 (Special
Observation Period 2) field campaign took place from 15 July to 05 August 2013 in the
western Mediterranean basin, at Ersa a remote site in Cape Corsica. During the campaign
more than 80 Volatile Organic Compounds (VOCs), including oxygenated species were
measured by different online and offline techniques. At the same time an exhaustive
description of the chemical composition of fine aerosols was performed. First we combined a
back-trajectory analysis and an estimation of photochemical age to characterize air mass
origins and chemical processing times, which confirmed the remote nature of the site.
Therefore, low levels of anthropogenic VOCs (typically tens to hundreds of ppt for individual
species) and black carbon (0.1-0.9 µg m$^{-3}$) were observed while significant levels of biogenic
species (peaking at ppb level) were measured. Furthermore, secondary oxygenated VOCs
(OVOCs) largely dominated the VOC speciation during the campaign, while Organic Matter
(OM) dominated the aerosol chemical composition (55% of the total mass of non-refractory
submicron aerosol on average).
Second, Positive Matrix Factorization (PMF) and Concentration Field (CF) analyses
were performed on a database containing 42 VOCs (or grouped VOCs), including OVOCs, to
identify co-variation factors of compounds that are representative of primary emissions, or
chemical transformation processes. A six-factor solution was found for the PMF analysis,
including a primary and secondary biogenic factor, both correlated to temperature and
exhibiting a clear diurnal profile. In addition, three anthropogenic factors characterized by
compounds of various lifetimes and/or sources have been identified (long-lived, medium-
lived and short-lived anthropogenic factors). The anthropogenic nature of these factors was
confirmed by the CF analysis which identified potential source areas known for intense
anthropogenic emissions (north of Italy and south-east of France). Finally, a factor
characterized by OVOCs of both biogenic and anthropogenic origins was found. This factor
was well correlated to submicron organic aerosols (OA) measured by an Aerosol Chemical
Speciation Monitor (ACSM) highlighting the close link between OVOCs and organic aerosols
measured at Cape Corsica mainly associated (96%) to secondary fraction of OA. The source
apportionment of OA measured by ACSM led to a 3-factor solution identified as Hydrogen-
like OA (HOA), Semi-Volatile-Oxygenated OA (SV-OOA) and Low-Volatile OOA (LV-
OOA) for averaged mass concentration of 0.13, 1.59, and 1.92 µg m$^{-3}$, respectively.
A combined analysis of gaseous PMF factors with inorganic and organic fractions of
aerosols helped distinguishing between anthropogenic/continental and biogenic influences on
the aerosol and gas phase compositions.



## 1 Introduction

Organic matter is directly emitted in the atmosphere both in the gas phase as Volatile Organic Compounds (VOCs) and in the aerosol phase as Primary Organic Aerosol (POA). The sources can be of biogenic (from land or marine ecosystems) and anthropogenic (from traffic, industrial activities or residential heating) origins. Once emitted, it can be transported over long distances and undergo chemical transformations due to atmospheric photo-oxidants such as ozone ($O_3$), the hydroxyl radical (OH) and the nitrate radical ($NO_3$) at night. The hydroxyl radical is the main oxidant in the atmosphere and, therefore, controls the fate of most VOCs through oxidation cycles that lead to the formation of tropospheric $O_3$ (Seinfeld and Pandis, 1998) and of a large number of secondary Oxygenated VOCs (OVOCs) (Atkinson et al., 2000, Goldstein and Galbally, 2007). OVOCs subsequently react with atmospheric oxidants leading to multi-functionalized compounds of lower volatility through a multigenerational oxidation process (Kroll and Seinfeld, 2008; Jimenez et al., 2009, Aumont et al., 2012). These semi-volatile compounds take part in the formation of Secondary Organic Aerosols (SOA) by condensation onto pre-existing particles (Kanakidou et al., 2005). Organic aerosols are of particular interest owing to their impact on human health (Pope and Dockery, 2006) and their direct (Forster et al., 2007) or indirect (Lohmann and Feichter, 2005) effect on earth's climate. Furthermore, chemical models suggest that the secondary organic gaseous fraction, still reactive and multi-functionalized several days after emission, can be transported over long distances, affecting the oxidant budget as well as the formation of ozone and SOA, at remote locations (Aumont et al., 2005; Madronich, 2006). It is therefore essential to understand the sources and fate of organic matter in the atmosphere, and especially its evolution during long range transport.

Positive Matrix Factorization (PMF) models (Paatero and Tapper, 1994, Paatero, 1997) have been widely used to identify and quantify sources of VOCs, generally in urban environments (e.g. Latella et al., 2005, Leuchner and Rappenglück, 2010, Gaimoz et al., 2011, Yuan et al., 2012). This type of analysis allows the separation of different sources (e.g. vehicular exhaust, fuel evaporation, residential heating etc…) and the apportionment of those sources to the VOC budget. PMF was also used at remote sites (Lanz et al., 2009, Sauvage et al., 2009, Leuchner et al., 2015), despite the need of assuming mass conservation between the source location and the measurement site in this approach (Hopke et al., 2003). In such environments, PMF can be used as a tool to identify aged primary sources as well as



photochemical formation of organic trace gases. This approach can, therefore, be useful to get
insights into the sources and processes involved in the evolution of organic trace gases
measured at remote locations. For example, Leuchner et al. (2015) applied the PMF to 24 $C_2$-
$C_8$ Non Methane HydroCarbons (NMHCs) measured at a remote site at Hohenpeissenberg
(980 m asl). These authors obtained 6 different factors assigned to primary biogenic
emissions, short-lived combustion, short- and long-lived evaporative emissions, residential
heating, and a background component.

8       Similar PMF approaches are also conducted on the organic fraction of aerosols

measured mostly by Aerosol Mass Spectrometers (AMS) to identify different components
characterized by their sources, their way of formation and/or their chemical composition (Ng
et al., 2010a, Zhang et al., 2011). For example, aerosol factors such as HOA (Hydrocarbon-
like Organic Aerosol), and OOA (Oxygenated-like Organic Aerosol) are commonly extracted
from AMS spectra using PMF analysis and are attributed to POA and SOA respectively
(Zhang et al., 2011). The latter can also be separated into several factors as a function of the
volatility: Low-Volatile-OOA (LV-OOA) and Semi-Volatile-OOA (SV-OOA) (Zhang et al.,
2011). For example, Hildebrandt et al. (2010) detected two types of OOA with low volatility
using PMF on AMS data recorded at Finokalia, an eastern Mediterranean remote site, while
no HOA was present in detectable amounts. On the contrary, PMF analysis applied on aerosol
measurements performed at an urban background site in Barcelona in spring, in the western
Mediterranean basin, revealed a significant impact of local primary emissions with
Hydrocarbon Organic Aerosol, Cooking Organic Aerosol and Biomass Burning Organic
Aerosol factors accounting for 44% of OA, although regional and local secondary sources
(LV-OOA and SV-OOA) dominated the OA burden (Mohr et al., 2012). Another study,
combining ACSM measurements and $^{14}$C analysis, conducted in Barcelona in summer 2013,
revealed a large contribution of anthropogenic sources for this environment with fossil OC
representing 46% to 57% of total OA. However, a larger contribution of secondary origin for
fossil OC (>70%) and non-fossil OC (37-60%) was observed, leading to a large fraction of
OA contained in OOA factors (Minguillon et al., 2016). Macro-tracer analysis represents an
alternative solution to apportion OA and can be used to allocate/verify specific OA factors
derived from PMF analysis. For instance, in atmospheres not impacted by biomass burning,
water-soluble organic compounds (WSOC) have shown to provide valuable information on
SOA that could be mainly of biogenic origin (Sullivan et al., 2004, 2006; Heald et al., 2006;
Miyazaki et al., 2006; Kondo et al., 2007; Weber et al., 2007; Hennigan et al., 2008b).



More recently, combined source apportionments of organic aerosol and VOCs were
performed in urban environments (Slowik et al., 2010; Crippa et al., 2013a), allowing a better
classification of organic aerosol (OA) from the PMF analysis. This type of analysis also
allowed getting insights into OA sources such as the identification of gaseous precursors.
Residential time analysis allow the geographical location of potential source areas by
combining measured or estimated variables at a receptor site with back-trajectory analyses
(Ashbaugh et al., 1985; Seibert et al., 1994; Stohl, 1996). Combined with PMF results, these
models have been used to locate source regions of the PMF factors (Hwang and Hopke, 2007;
Lanz et al., 2009; Tian et al., 2013). This association of receptor-oriented models can be
powerful to identify the nature of the source or the chemical processes characterizing the PMF
factors. The Concentration Field (CF) is one of these source-receptor inverse models, which
was developed by Seibert et al. (1994). It consists in a redistribution of the measured or
estimated variables in grid cells along estimated back-trajectories.
The Mediterranean basin is an ideal location to study the sources and the fate of organic
carbon during long range transport since it is impacted by strong natural and anthropogenic
emissions and undergoes intense photochemical events (Lelieveld et al., 2002). The
ChArMEx project (Chemistry-Aerosol Mediterranean Experiment) aims at assessing the
present and future state of the atmospheric environment and of its impacts in the
Mediterranean basin. This initiative proposes to set up a coordinated experimental effort for
an assessment of the regional budgets of tropospheric trace species, of their trends, and of
their impacts on air quality, marine biogeochemistry, and regional climate. For that purpose an
intensive field campaign was performed during the summer 2013, at Cape Corsica (North of
Corsica Island) where a full suite of trace gases and aerosol species were measured for 3
weeks. In the framework of ChArMEx, the CARBOSOR (CARBOn within continental
pollution plumes: SOurces and Reactivity) project aimed more specifically at investigating the
sources of primary and secondary organic trace gases, as well as the composition of
continental plumes reaching Cape Corsica, with the goal of assessing their impacts on the
photo-oxidants and/or SOA sources and levels.
As part of the ChArMEx and CARBOSOR projects, this study investigates the
composition, the sources and the chemistry of atmospheric organic matter by combining
different statistical tools, i.e. the PMF, ME-2 (Multilinear Engine-2) models and the
Concentration Field method. This approach was used to (i) identify co-variation factors of
measured VOCs that are representative of primary emissions at various stages of ageing and
chemical transformations occurring during long range transport, and to (ii) better characterize



the different fractions of organic aerosol. The PMF factors were then used to assess the origin
of non-refractive organic species in PM$_1$ (Particulate Matter with aerodynamic diameter
below 1 μm) observed at the measurement site, and especially to try to determine the fraction
of biogenic versus anthropogenic OA.
**2 The ChArMEx experiment**
**2.1 Description of the Cape Corsica ground site**
The ChArMEx SOP2 (Short Observation Period 2) field campaign took place from 15
July to 05 August 2013. The measurement site is located at Ersa in Cape Corsica (42.969°N,
9.380°E), at the top of a hill (alt 533 m above sea level, asl), a few kilometres away from the
sea in all directions (6, 4.5, and 2.5 km from the east, north and west sides respectively) (see
Figure 1). The measurement site is surrounded by widespread vegetation such as "maquis"
shrub-land typical of Mediterranean areas (Zannoni et al., 2015). The closest city, Bastia, is
located approximately 30 km south of the site. It is the second largest city in Corsica (44 121
inhabitants, census 2012), which hosts the main harbour of the island with about 413 000 and
614 000 passengers in July and August 2013, respectively (CCI Territorial Bastia Haute
Corse, 2013). However, the Cape Corsica peninsula is characterized by a mountain range
(peaking between 1,000 and 1,500m asl), which acts as a natural barrier isolating the
measurement site from any atmospheric flow originating from Bastia.
**2.2 VOC measurements**
During the ChArMEx SOP2 field campaign more than 80 VOCs, including Non-
Methane HydroCarbons (NMHCs) and Oxygenated (O)VOCs, were measured using
complementary online and offline techniques whose sampling inlets were located
approximately 1.5 m above the roof of a trailer where the instruments were housed. Table 1
summarizes the VOCs measurement performed during the campaign.
Sixteen (16) protonated masses were extracted from the Proton Transfer Reaction-
Time of Flight Mass Spectrometer (PTR-ToF-MS, KORE Inc® 2$^{nd}$ generation) leading to the
measurements of OVOCs (alcohols such as methanol: m/z 33.03, aldehydes and ketones,
carboxylic acids), aromatics (sum of both C-8 and C-9 aromatics: m/z 107.09 and m/z 121.10,
respectively) and biogenic VOCs (BVOCs such as isoprene: m/z 69.07 and the sum of



monoterpenes: m/z 137.13). Ambient air was sampled through a 5-m long PFA line (PerFluroAlkoxy, 1/4"-o.d.) held at 50°C using a constant flow rate of 1.2 L min$^{-1}$ to minimize the residence time to 4 s. The PTR-ToF-MS sampling flow rate was set at 150 mL min$^{-1}$ and an additional pump was used to raise the flow rate to the required 1.2 L min$^{-1}$ in the sampling line. The instrument was operated at reactor pressure and temperature of 1.33 mbar and 40°C, respectively, leading to an E/N ratio of 135 Td.

An automated zero procedure was performed every hour for 10 min. Humid zero air was generated by passing ambient air through a catalytic converter (stainless steel tubing filled with Pt wool held at 350°C) allowing to get the same relative humidity than in ambient air. During the campaign, the PTR-ToF-MS was calibrated every three days using a Gas Calibration Unit (IONICON®) and various standards including a mix of 15 VOCs (including NMHCs, OVOCs and Chlorinated VOCs) in a canister (Restek®), a mix of 9 NMHCs in a second cylinder (Praxair®) and a mix of 9 OVOCs in a cylinder (Praxair®) (see supplementary material Table S1). Additional calibrations were performed before and after the campaign using permeation tubes (Kin-Tec Inc®) for carboxylic acids and a Liquid Calibration Unit (IONICON®) with a certified solution for methylglyoxal. To account for a possible drift of the PTR-ToF-MS sensitivity during the campaign, relative calibration factors were determined for the carboxylic acids and methyl glyoxal using a specific VOC as a reference (present in the standard mixtures used to calibrate the PTR-ToFMS during the campaign and with a m/z value as close as possible for each compound: e.g. acetaldehyde, acetone and methylethylketone for formic acid, acetic acid, and methylglyoxal, respectively).

Signal of every unit mass is accumulated over 10 min and normalized by the signals of $H_3O^+$ and the first water cluster $H_3O^+(H_2O)$ as proposed by de Gouw and Warneke (2007). Concentrations are calculated using Eq. (1):

$$[X] = \frac{i_{X\_net}}{(i_{H_3O^+} + X_r . i_{H_3O^+(H_2O)})} \cdot \frac{150000}{R_{f,X}} \tag{1}$$

Where [X] represents the mixing ratio of a given VOC, $i_{X\_net}$ the net signal recorded for this VOC, $i_{H_3O^+}$ and $i_{H_3O^+(H_2O)}$ the signals of $H_3O^+$ and $H_3O^+(H_2O)$ at m/z 19 and 37, but respectively recorded at m/z 21 and 39, to avoid any saturation of the detector at m/z 19 and 37, and recalculated using the isotopic ratio between $^{16}O$ and $^{18}O$. $X_r$ is a factor introduced to account for the effect of humidity on the PTR-ToFMS sensitivity (de Gouw and Warneke, 2007) and determined experimentally through calibrations performed at various relative humidities. $R_{f,X}$ is the sensitivity determined during calibration experiments (in ncts ppt$^{-1}$) and



normalized to 150000 counts s$^{-1}$ of $H_3O^+$ ions. The latter is the number of counts of reagent
ions observed in our PTR-ToFMS instrument.
Forty-three (43) $C_2$-$C_{12}$ NMHCs, including alkanes, alkenes, alkynes and aromatics,
were measured using an online Gas Chromatograph (GC) equipped with two columns and a
dual Flame Ionization Detection (FID-FID) system (Perkin Elmer®). This instrument has been
previously described in detail by Badol et al. (2004). Briefly, air is sampled via a 5-m length
PFA line (1/8"-o.d.) at a flow rate of 15 mL min$^{-1}$. Ambient air passes through a Nafion
membrane to dry it and is then pre-concentrated during 40 min onto a sorbent trap made of
Carbopack B and Carbosieve SIII and held at -30°C by a Peltier cooling system. The trap is
then heated up to 300 °C (40 °C s$^{-1}$) to desorb and inject VOCs in a Perkin Elmer GC system.
The chromatographic separation is performed using two capillary columns thanks to a
switching facility. This approach allows for a better separation and reduces co-elution issues
(Badol et al., 2004). The first column designed for $C_6$-$C_{12}$ compounds is a CP Sil 5CB
(50 m×0.25 mm×1 μm), while the second column designed for the $C_2$-$C_5$ compounds is a plot
$Al_2O_3$/$Na_2SO_4$ (50 m×0.32 mm×5 μm). The separation step lasts 50 min, leading to a total
time resolution of 1h30'. Finally, eluted compounds are detected using the 2 FID detectors.
Calibrations were performed at the beginning, at the middle and at the end of the campaign,
using a standard mixture containing 32 compounds (NPL®, see supplementary material Table
S2).
Sixteen (16) $C_3$-$C_7$ OVOCs, including aldehydes, ketones, alcohols, ethers, esters, as
well as 6 NMHCs, including BVOCs and aromatics, were measured using an online GC/FID-
Mass Spectrometer (MS). This instrument has been described in details by Roukos et al.
(2009). Ambient air is sampled via a 5-m length PFA line (1/8") at a flow rate of 15 mL min$^{-1}$
by an air server unit (Markes International®, Unity I) and passes through a KI ozone scrubber.
The sampled air is pre-diluted (50% dilution) with dry zero air to keep the relative humidity
below 50%. A sample is then collected into an internal trap cooled by a Peltier system at
12.5 °C and consists in a 1.9 mm i.d. quartz tube filled with two different sorbents (5 mg of
Carbopack B and 75 mg of Carbopack X, Supelco®). Compounds trapped onto the sorbents
are then thermally desorbed at 280 °C and injected into the column and analyzed by a GC
(Agilent®) equipped with a FID for quantification and a Mass Spectrometer (MS) to help with
the identification. The thermodesorbed compounds are passed through a high polar CP-lowox
column (30 m×0.53 mm× 10 μm, Varian®) for separation. The sampling and analysis steps
last 40 and 50 min, respectively, for a total time resolution of 1h30'. Calibrations were





performed several times during the campaign, using a standard mixture containing 29
compounds (Praxair®, see supplementary material Table S2).
Thirty-five (35) $C_5$-$C_{16}$ NMHCs, including alkanes, alkenes, aromatics and BVOCs, as
well as 5 $C_6$-$C_{12}$ n-aldehydes were collected by active sampling into sorbent cartridges using
an Automatic Clean Room Sampling System (ACROSS-TERA Environment®) and were
analyzed later by GC-FID. This technique has already been described by Detournay et al.
(2011) and its set-up in the field was discussed by Ait-Helal et al. (2014). Briefly, air was
sampled via a 3-m length PFA line (1/4"-o.d.) at 200 mL min$^{-1}$ and was passed through a
$MnO_2$ ozone scrubber and a stainless-steel particle filter (2μm pore size diameter). VOCs are
collected during 3 h in cartridges filled with Carbopack C (200 mg) and Carbopack B
(200 mg), formerly conditioned with purified air at 250 °C during 24 h.
Finally, sixteen (16) $C_1$-$C_8$ carbonyl compounds were collected offline during 3 h using
the same sampling device than for solid-sorbent cartridges by active sampling on
DiNitroPhenylHydrazine (DNPH) cartridges (Waters®). These compounds were analyzed
later by High Performance Liquid Chromatography (HPLC) with UV detection. Air was
sampled via a 3-m length PFA line (1/4"-o.d.) at 1.5 L min$^{-1}$ and was passed through a KI
ozone scrubber and a stainless-steel particle filter (2μm pore size diameter). Data are available
only for the first 10 days of the campaign (15/07-25/07) due to unresolved leakage issues for
the rest of the campaign and hence a contamination of the cartridges with indoor air from the
trailer was suspected.
The detection limits for each species measured by all five techniques were determined
as 3σ of the blank variation for PTR-ToF-MS and offline sampling methods and as 3σ of the
baseline fluctuations for online GCs. The uncertainties for each species were estimated
following the "Aerosols, Clouds, and Trace gases Research InfraStructure network"
(ACTRIS) guidelines for uncertainty evaluation (ACTRIS Measurement Guideline VOC,
2012), taking into account precision, detection limit and systematic errors of the
measurements. The range of uncertainties and detection limits for each technique is given in
Table 1. Furthermore, systematic intercomparisons for compounds measured by different
techniques (e.g. isoprene, monoterpenes, acetone, n-pentane, benzene, etc…) were performed
to validate the database (not shown).

**32    2.3 Ancillary gas measurements**





During the campaign, measurements of other trace gases (NO, $NO_2$, $O_3$, CO, $CO_2$, $CH_4$,
$H_2O$, $SO_2$) were additionally performed at the same measurement site.
NO and $NO_2$ were measured by a commercial analyzer (CRANOX II, EcoPhysics®)
using ozone chemiluminescence with a time resolution of 5 min. Since this technique allows
the direct measurements of NO only, $NO_2$ was converted into NO using a photolytic converter
incorporated in the analyzer.
$O_3$ was measured using a UV absorption analyzer (TEI 49i, Thermo Environmental
Instrument Inc®) at a time resolution of 5 min. CO, $CO_2$, $CH_4$, $H_2O$ were simultaneously
measured by a commercial analyzer (G2401, PICARRO®) based on Cavity Ring Down
Spectroscopy (CRDS). Finally, $SO_2$ was measured by a commercial analyzer (TEI 43i,
Thermo Environmental Instrument Inc®) using fluorescence spectroscopy at a time resolution
of 5 min.
**2.4 Aerosol measurements**
Online measurements of organic aerosols (PILS-IC, PILS-TOC, OCEC Sunset Field
Instruments, Q-ACSM) have been available since beginning of June 2013, but data reported
here are restricted to the ChArMEx SOP2 period (15/07-05/08/2013) for which VOC
measurements have been performed.
In addition, Black carbon (BC) has been continuously monitored, during the same
extended period, using a 7-wavelength aethalometer model AE-31 (MAGEE Scientific®) at a
time resolution of 15 min.
**2.4.1. PILS-IC instrument**
Measurements of major anions ($Cl^-$, $NO_3^-$, $SO_4^{2-}$), cations ($Na^+$, $NH_4^+$, $K^+$, $Mg^{2+}$, $Ca^{2+}$)
and light organics (methanesulfonate (MSA), oxalate) in $PM_{10}$ were performed using a
Particle-into-Liquid-Sampler (PILS; Orsini et al., 2003) running at $11.8\pm0.5$ L $min^{-1}$ and
coupled with two Ion Chromatographs (IC). More details on the settings of the PILS-IC can
be found in Sciare et al. (2011). During this field campaign, ambient concentrations of ions
were corrected from blanks performed every day for 1h and achieved by placing a total filter
upstream of the sampling system. Very low blank values (typically below 1 ppb) were
systematically detected for all ions providing further confidence on the efficiency of the
acidic/basic denuders set upstream of the PILS, the lack of contaminants in our system, and



the quality of our Milli-Q water during the whole duration of the study. Liquid flow rates of
the PILS were delivered by peristaltic pumps and set to 1.0 ml min$^{-1}$ for producing steam
inside the PILS and 0.37±0.02 ml min$^{-1}$ for rinsing the impactor. Calibrations (5 to 7 points)
of anions and cations (including light organics) were performed every two weeks (from end of
May 2013 to beginning of August 2013) with no significant drift reported (e.g. below 5%
difference on average). Based on IC settings, the detection limit (2σ) for ions was typically
0.1 ppb, which corresponds to an atmospheric concentration of ~1 ng m$^{-3}$. The overall
uncertainty associated with PILS-IC measurements includes variability in air sampling flow
rate, liquid flow rate, calibration, and collection efficiency and was estimated to be of the
order of 25%. Time resolutions were typically of 24 min for anions (including light organics)
and 12 min for cations. Because this study focuses on organics in the atmosphere, only MSA
and oxalate data will be presented and discussed here. A total of 761 and 996 valid data points
of MSA and Oxalate were obtained, respectively, with concentrations ranging for MSA from
4 to 59 ng m$^{-3}$ (21 ng m$^{-3}$ on average) and ranging for oxalate from 1 to 24 ng m$^{-3}$ (10 ng m$^{-3}$
on average).
**2.4.2. PILS-TOC instrument**

19       Measurements of water-soluble organic compounds (WSOC) in PM$_1$ were performed

every 4 min using a modified Particle-into-Liquid-Sampler (Brechtel Manufacturing Inc.,
USA; Sorooshian et al., 2006) coupled with a total organic carbon analyzer (TOC, Model
Sievers 900, Ionics Ltd, USA). More information on the operation procedure of this
instrument is provided by Sciare et al. (2011). Briefly, the PILS-TOC instrument is running at
15 L min$^{-1}$ and a measured dilution factor of 1.30 was taken in the instrument which is close
to the one reported by Sullivan et al. (2006). A polyethylene filter of 0.45 μm pore size
diameter was set in-line in the aerosol liquid flow (downstream of the PILS collector) in order
to analyze solely the water-soluble OC fraction. Daily blanks for the PILS-TOC instrument
were achieved by placing a total filter upstream of the sampling system for 1h. In this
configuration, approximately 15 min were necessary to reach blank values which were very
stable during the campaign showing a mean concentration of 35.6±2.6 ppbC, very similar to
the one reported by Sciare et al. (2011). Note that most of the blank concentration refers to the
TOC concentration in the ultra-pure water used in the PILS instrument (typically 25 ppbC),
suggesting little contamination in the PILS instrument as well as a good efficiency of the
VOC denuder placed upstream. Note also that the daily blanks for the PILS-TOC instrument



were performed at different hours of the day and did not show a clear diurnal pattern that
could be linked to diurnal variations of VOCs. Ambient WSOC measurements were then
corrected from this blank value. Limit of quantification of ambient WSOC measurements was
estimated as $2\sigma$ (twice the uncertainty calculated for the blank concentrations), corresponding
to about 0.48 µgC m$^{-3}$. A total of 6592 valid data points were collected during the period of
the study (15/07-05/08/2013), corresponding to a mean ambient (blank corrected) WSOC
concentration of 11.6±6.7 ppbC (i.e. 1.00±0.60 µgC m$^{-3}$).

### 9   2.4.3. OCEC Sunset Field instrument

Semi-continuous (2-h time resolution) concentrations of elemental carbon (EC) and
organic carbon (OC) in PM$_{2.5}$ were obtained in the field from an OCEC Sunset field
instrument (Sunset Laboratory, Forest Grove, OR, USA; Bae et al., 2004) running at 8 L min$^{-}$
$^{1}$. A denuder provided by the manufacturer was set upstream in order to remove possible
adsorption of VOCs onto the filter used to collect fine aerosols in the instrument.
Measurement uncertainty given by the OCEC Sunset field instrument is poorly described in
literature and an estimate of 20% for this uncertainty was taken here following Peltier et al.
(2007). This instrument has been running continuously for the whole duration of the
campaign (15/07-05/08/2013) with 252 valid EC and OC data points obtained.

### 21   2.4.4. Q-ACSM instrument

Since summer 2012, measurements of the chemical composition of non-refractory
submicron aerosol (NR-PM$_1$) have been carried out at the measurement site using a
Quadrupole Aerosol Chemical Speciation Monitor (Q-ACSM, Aerodyne Research Inc.
Billerica, MA). This recently developed instrument shares the same general structure with the
Aerosol Mass Spectrometer (AMS) but has been specifically developed for long-term
monitoring. An exhaustive description of ACSM is available in Ng et al. (2011) while a
growing number of studies have already reported long-term observations of NR-PM$_1$
composition and concentrations using it (Ripoll et al., 2015; Minguillón et al., 2015; Petit et
al., 2015; Parworth et al., 2015; Budisulistiorini et al., 2015).
The Q-ACSM instrument used here participated to the large intercomparison study of
13 Q-ACSM that took place at the ACMCC (Aerosol Chemical Monitor Calibration Center;
https://acmcc.lsce.ipsl.fr/), three months after this field campaign and showed - for




atmospheric concentrations and fragmentation pattern - very consistent results in terms of reproducibility and consistency (Crenn et al., 2015). Source apportionment performed with the same Q-ACSM (during the intercomparison study at ACMCC) has also led to very consistent and comparable results (Frölich et al., 2015). The calibration of this instrument with mono-dispersed (300 nm diameter) ammonium nitrate particles was performed at ACMCC in May 2013, about two months before the start of this study. Because ambient air was dried by a Nafion membrane before entering into the Q-ACSM and because ammonium nitrate was not significant during the field campaign, we have kept here a constant collection efficiency (CE) of 0.5. Onsite atmospheric concentrations delivered by the Q-ACSM were consistent for NR-PM$_1$ and SO$_4$ concentrations obtained with co-located online instruments (Scanning Mobility Particle Sizer and Particle-Into-Liquid-Sampled-Ion-Chromatograph). The Q-ACSM instrument has been continuously operating for the whole duration of the campaign (15/07-05/08/2013), with a total of 1148 valid data points of 30 min time resolution each.

## 2.5 Back-trajectory classification

A study of back-trajectories was performed to identify and classify the origin and typology of the different air masses reaching Cape Corsica during the campaign and to support interpretation of the results. Back-trajectories of 48 h were calculated, every 6 h during the whole campaign, with an ending point at the measurement site (42.969°N, 9.380°E, alt: 600 m asl) using the online version of the HYSPLIT (HYbrid Single-Particle Lagrangian Integrated Trajectory) model developed by the National Oceanic and Atmosphere Administration (NOAA) Air Resources Laboratory (ARL) (Draxler and Hess, 1998; Stein et al., 2015). This model was chosen for its easy and quick visualisation facility.

A visual classification of these back-trajectories was performed as a function of their origin, altitude and wind speed and segregated into five clusters (Figure 2). A description of the 5 clusters is provided in Table 2. Four clusters correspond to different wind sectors defined by the origin of the air masses reaching the measurement site (West, North-East, South and North-West). These clusters are characterized by different transit times since the last potential anthropogenic contamination (i.e. since the air mass left the continental coasts). Indeed, the air masses from the "Marine-West" cluster have spent 36 to more than 48 h above the sea, while they have spent 10-20 h and 12-18 h for the "Europe-North-East", and the "France-North-West" clusters respectively (Table 2). For the "Corsica-South" cluster, the





indicated transit time (12-24 h) considers the time spent by air masses above land (Corsica
and Sardinia Islands) before passing over the sea. These different transit times potentially
indicate different atmospheric processing times for the air masses, the longest being for the
"Marine-West" cluster.

5        The last cluster gathers air masses transported over short distances over 48 h and

therefore during calm situations with low wind speed (Figure 2). The "Calm-Low Wind"
cluster and the "Marine-West" cluster are the two most representative clusters, representing
each 30% of the air masses origin. They are followed by the "Europe-North-East" cluster
representing 26%, and then by the "Corsica-South" and "France-North West" representing
8% and 6% of the air mass origins, respectively.
**2.6 Photochemical age of air masses**

14        Regarding the relative long transit time of air masses travelling from continental source

areas to the measurement site (from 10 to more than 48 h, see section 2.5), the assessment of
the photochemical age using the field observations can be performed with specific ratios of
long-lived VOCs measured at significant levels at the site. The use of graphic representations
of the ratios of three different alkanes, such as ln(butane/ethane) vs ln(propane/ethane) is well
suited to assess the photochemical age of air masses which experienced long-range transport
(Rudolph and Johnen, 1990; Jobson et al., 1994; Parrish et al., 2007). Considering an air
parcel isolated from any new emission or mixing with other air parcels, and considering that
the main loss of alkanes is their oxidation by the OH radical; one can estimate the relation of
three alkanes as described by eq. (2) (Jobson et al., 1994).

$$Ln\frac{[bu\tan e]}{[ethane]} = \frac{k_{bu\tan e} - k_{ethane}}{k_{propane} - k_{ethane}} Ln\frac{[propane]}{[ethane]} + \beta \qquad (2)$$

$k_i$ is the bimolecular reaction rate constant of the reaction between the specie i and OH.
The $\beta$ parameter depends on the emission ratios of these three species and the reaction rate
constants.

27        Since ethane is the less reactive of these compounds, the ratios will tend to decrease

with increasing photochemical age. The evolution of Ln(butane/ethane) as a function of
Ln(propane/ethane) during the ChArMEx SOP2 field campaign in Cape Corsica is presented
in Figure 3. The points of Figure 3 have been colour-coded as a function of the back-
trajectory clusters described in the previous section.



Figure 3 reveals that the air masses of the Marine West (light blue) cluster present
higher photochemical ages (lower alkane ratios) relatively to the air masses of the European-
North-East (purple) cluster, consistent with the analysis of back-trajectories (c.f. section 2.5).
Moreover, the good linearity observed in the evolution of the ratios allows the qualitative
comparison of the photochemical age of air masses from the different wind clusters.

6       These ratios have been compared to ratios observed at measurement sites of different

types (see Supplementary Material Fig. S3). The ratios obtained during the campaign cover a
large range of values with, in particular, very low values for the Marine-West cluster, typical
of relatively aged air masses sampled at very remote sites. It indicates that air masses can
spend several days over the sea before reaching the measurement site especially for the
Marine-West cluster. In general, ratios representative of remote locations are observed all
along the campaign, confirming the remote nature of the Cape Corsica station.
It is noteworthy that the slope observed for our dataset (0.65, see Figure 3) is
significantly lower than the theoretical ones calculated for an isolated air mass experiencing a
selective oxidation by OH (2.50) or by Cl (1.97). The lack of concordance with theoretical
slopes has often been observed (e.g. Parrish et al., 1992; McKeen et al., 1996) and has been
attributed to the mixing between air parcels of different histories and origins during long-
range transport (Parrish et al., 2007 and references therein).
**3 Source-receptor models**
**3.1 The Positive Matrix Factorization (PMF)**
In this study, US E.P.A PMF 3.0 was used to perform the factor analysis. For a
detailed presentation of the PMF principle, the reader can refer to the first description made
by Paatero and Tapper (1994) and to the user's guide written by Hopke (2000). Briefly, a
specific dataset at a receptor site can be viewed as a data matrix X containing i samples and j
measured chemical species. The PMF identifies the number of factors p, i.e. the number of
emission sources and/or chemical processes, driving ambient concentrations of the measured
species. It, therefore, allows decomposing the matrix X into a product of two matrices: the
species profile (f) of each source with a dimension of p×j (representing the repartition of each
measured chemical species in the factors); the contribution (g) of each factor to each sample
with a dimension of i×p (representing the time evolution of each factor); and allows
minimizing the residual error e. This is summarized in eq. (3):





$$X_{ij} = \sum_{k=1}^{p} g_{ik} \times f_{kj} + e_{ij} \tag{3}$$

1        The minimization of the residual sum of squares Q is performed using eq. (4) to derive

the solution of eq. (3).

$$Q = \sum_{i=1}^{n} \sum_{j=1}^{m} \frac{e_{ij}^2}{s_{ij}^2} = \sum_{i=1}^{n} \sum_{j=1}^{m} \left[ \frac{X_{ij} - \sum_{k=1}^{p} g_{ik} \times f_{kj}}{s_{ij}} \right]^2 \tag{4}$$

3        Where $S_{ij}$ is the uncertainty matrix associated to the data matrix $X_{ij}$, estimated as

described in section 2.2.

5        The PMF analysis was conducted on a dataset of 42 species, including NMHCs, and

OVOCs measured by the two online GCs and the PTR-ToFMS, and 329 observations, the
time resolution being 1h30' (time resolution of the GCs). Measurements by active sampling
on sorbent and DNPH cartridges were not included in this dataset due to their low time
resolution (3h), which would have resulted in too few observations. Furthermore, compounds
were not considered when missing, when more than half of observations were below the
detection limit, or when associated to a low signal-to-noise ratio (s/n < 1 in our case). Missing
values and values below the detection limit in the selected dataset were replaced by the
geometric mean and half of the detection limit, respectively, following the method used by
Sauvage et al. (2009). To minimize the weight of these observations in the PMF results, the
uncertainties of missing values and values below the detection limit were set to 4 times the
geometric mean and 5/6 of the detection limit, respectively. PMF also allows the
minimization of the contribution of species of low signal-to-noise ratio (s/n < 1.5 in our case)
by declaring these species as "weak" and hence tripling their original uncertainties. Fourteen
species have been declared as "weak" in this work.

20        Ethane, methanol and acetone are characterized by high background concentrations at

the measurement site. To minimize the weight of these three species in the PMF results, their
estimated background concentrations (500, 1000 and 1200 ppt for ethane, methanol and
acetone, respectively) were subtracted to the measured concentrations in the data matrix X.
These values were chosen arbitrarily to subtract the background concentrations of these
species keeping their variability and avoiding near zero values.

26        The PMF was run following the protocol proposed by Sauvage et al. (2009) and lying

on several statistical indicators (unexplained part for each factors, correlation between the
sum of the factor contributions and the sum of the measured concentration, the parameter Q
(see above), mean and standard deviation of scaled residuals …) to determine the optimal





model parameters (number of factors, rotational parameter Fpeak) leading to the best solution.
Based on this approach, we have derived a final solution with 6 factors for a Fpeak of -0.5.
Moreover, the homogeneity of the database built using measurements from different
techniques was studied to ensure that all instruments are well-represented in the solutions.
This was done by checking that no substantial differences are observed between the scale
residuals of the different instruments. We therefore calculated the mean of the absolute values
of scaled residuals for the three instruments $\overline{\left(\dfrac{|e_{ij}|}{s_{ij}}\right)}$ (0.73, 0.67 and 0.75 for the PTR-ToF-MS,
GC/FID-FID and GC/FID-MS, respectively). The differences observed between these
parameters calculated for the three instruments are lower than 0.08. This indicates a
reasonable homogeneity of the instrument databases (concentrations, uncertainties) since
absolute differences below 0.25 have been determined to be satisfactory to avoid over-
weighting of the measurements of a particular instrument in PMF solutions (Crippa et al.,
2013a)). Therefore, no scaling procedure was performed on the database used in our PMF
analysis.
Furthermore, 100 bootstrap runs were performed for the 6 factors solution to estimate
the stability and uncertainty of this solution. This operation consisted in performing additional
PMF runs using new input data files built by randomly selecting non-overlapping blocks of
the original data matrix, the contribution of each factor derived from these runs being then
compared to the original solution. The lowest correlation coefficient between bootstrap
solutions and base run solutions was 0.6. The 6-factors solution appeared to be well-mapped
in the base run with mapping of bootstrap factors to base run factors higher than 86% for all
factors (see Supplementary material S4).
**3.2 Multilinear Engine (ME-2)**
Source apportionment of organic aerosol components from Q-ACSM was performed
using Positive Matrix Factorization (PMF, Paatero, 1997; Paatero and Tapper, 1994) via the
ME-2 solver (Paatero, 1999). An extended Q-ACSM dataset of 2 months (starting from 05/06
till 5/08/2013) was used here in order to obtain a wider range of atmospheric variability and
improve PMF output results. The extraction of OA data and error matrices as mass
concentrations in μg m$^{-3}$ over time, as well as their preparation for PMF/ME-2 according to
Ulbrich et al. (2009), was done within the ACSM software, except for the down weighing



procedure of mass fragments which was performed using the interface source finder (SoFi,
Canonaco et al., 2013), version 6.1. Only m/z up to 100 were considered here since they
represented nearly the whole OA mass (around 98 %) and did not interfere with ion fragments
originating from naphthalene. The interface SoFI was used to control ME-2 for the PMF runs
of the ACSM OA data. Unconstrained PMF runs were investigated here with 1 to 6 factors
and a moderate number of seeds (10) for each factor number without no conclusive results on
the consistency of mass spectra profile obtained for the different factors. Constrained PMF
runs have been investigated for that purpose with fixed factors for HOA (Hydrogen-like OA),
with much more conclusive results and significant improvements compared to the
unconstrained PMF. The results presented here were obtained using constrained PMF using
an averaged HOA profile taken from Ng et al. (2010b) and constrained with a value of 0.1.
The proper constrained PMF solution was selected based on the recommendations from
Canonaco et al. (2013) (e.g. consistency of the factor profiles mass spectra, consistency of
times series with external tracer, low Q/Qexp value). These criteria are presented and
discussed hereafter.
In this study, we therefore applied separate factorization analysis to both VOCs and
aerosol databases. Another approach consists in a factorization analysis of combined aerosol
and gaseous databases (Slowik et al., 2010; Crippa et al., 2013a). Thus, an attempt to perform
such PMF analysis was conducted, using the gaseous database (42 VOCs) described above
and full ACSM spectra as inputs and taking care of the homogeneity of the different inputs by
applying a scaling procedure as proposed by Slowik et al. (2010) and Crippa et al. (2013a).
However, it did not allow to satisfactorily apportion aerosol measurements and led to weaker
solutions than the ME-2 analysis. It was therefore decided to keep separated solutions for both
gas and aerosol phase organics.
**3.3 The Concentration Field model (CF)**
Receptor-oriented models have been developed to identify, localize and quantify
potential source areas which impact the concentrations of a variable measured at a receptor
site in the form of maps of a contribution quantity. In this study we have used the
Concentration Field (CF) approach developed by Seibert et al. (1994). This method consists in
redistributing concentrations of a variable observed at a receptor site along the back-
trajectories, ending at this site, inside a predefined grid (0.5° x 0.5°, for this study). The





calculated concentrations in each grid cells are weighted by the residence time that air parcels
spent in each cell following eq.

3  (5):

$$\log \overline{C}_{ij} = \frac{\sum_{L=1}^{M}(n_{ijL} \times \log C_L)}{\sum_{L=1}^{M} n_{ijL}} \tag{5}$$

Where $C_{ij}$ is the calculated concentration of the ij-th grid cell, L the back-trajectory
index, M the total number of back-trajectories, $C_L$ the concentration measured at the site when
the back-trajectory L reached it and $n_{ijL}$ the number of points of the back-trajectory L which
fall in the $ij^{th}$ grid cell. The latter is representative of the time spent by the back-trajectories in
the ij-th grid cell since a constant time step of 1 h is used between each point of a back-
trajectory.
The 3-day back-trajectories (selected to account for distant potential source areas of
species of long lifetimes), used in the CF analysis, were calculated by the British Atmospheric
Data Centre (BADC) model every hour. This model uses the wind fields calculated by the
European Centre for Medium-range Weather Forecasts (ECMWF) to determine the
trajectories of air masses. This model was selected here instead of Hysplit for convenience,
since format of output files matches the needed one for our CF model. Comparison of
randomly selected back-trajectories, in each identified clusters (see section 2.5), calculated by
both models (BADC and Hysplit) has revealed satisfactory agreement in terms of origin and
areas over-flown. The BADC back-trajectories were interrupted when the altitude of the air
mass exceeded 1500 m (asl), to get rid of the important dilution affecting air masses in the
free troposphere (the boundary layer height has been arbitrary set here to 1500 m (asl) for all
trajectories). Furthermore, the grid cells containing less than 5 trajectory points were not
considered for robustness purposes.
To take into account the uncertainties associated to the back-trajectories, a smoothing of
concentrations was applied to all the grid cells values as recommended by Charron et al.
(2000) and using eq. (6).

$$C_{ij-l} = \frac{\left(\sum_{p=1}^{8} C_p + C_{ij}\right)}{9} \tag{6}$$



Where $C_{ij-1}$ is the calculated concentration of the $ij^{th}$ grid cell after smoothing, $C_{ij}$ the
calculated concentration of the grid cell before smoothing and $C_p$ (1<p<8) the concentrations
before smoothing of the 8 neighbour grid cells.
**4 Results and Discussion**
**4.1 Overview of gaseous and aerosol measurement results**
**4.1.1. Gas phase**
The measured mixing ratios of some organics (acetylene, isoprene, sum of
monoterpenes, and acetone) as well as inorganic trace gases (CO, NO, $NO_2$, $O_3$) and wind
direction are presented in Figure 4. Anthropogenic long-lived species such as acetylene and
CO present similar temporal variations during the campaign. Indeed, we noticed a slow
variation of these compounds with a rise at the beginning of the campaign that reaches a
maximum on 21 July and a subsequent decrease. The maximum corresponds to a period when
air masses came from areas with strong emissions of anthropogenic species (North of Italy).
However, the rise observed the previous days did not correspond to specific air mass cluster.
Furthermore, the levels of anthropogenic species are very low at the measurement site (below
200 ppt for acetylene, also observed for other anthropogenic compounds: e.g. below 80, 120
and 150 ppt for benzene, n-butane and toluene, respectively) highlighting the probable lack of
local anthropogenic sources. These very low levels of anthropogenic species at the ground
level (often close to the limit of detection) made their measurements very challenging during
the campaign.
On the contrary, significant levels of primary biogenic compounds were observed and
could reached up to 1.2 and 2.0 ppb for isoprene and the sum of monoterpenes, respectively
(Figure 4). These compounds were locally emitted by the typical vegetation of the
Mediterranean region ("maquis" shrub-land) surrounding the measurement site. The mixing
ratios for these compounds present a clear diurnal cycle with the highest values coinciding
with maxima of temperature and solar radiation. Two periods characterized by high mixing
ratios of biogenic VOCs were observed (27-28 July and 02-04 August), which correspond to
the warmest periods of the campaign.
Oxygenated VOCs such as acetone were also present at significant levels: up to 3.8
ppb (Figure 4). This compound has primary and secondary sources, issued from oxidation of



both biogenic and anthropogenic VOCs (see discussion in section 4.2.3). Therefore, acetone levels increase both when anthropogenic VOC concentrations increase (first part of the campaign) and when intense biogenic emission are observed (27-28 July and 02-04 August).

$NO_x$ levels remained low (<0.5 and <2.0 ppb for NO and $NO_2$, respectively) during the whole campaign. This confirms the lack of local anthropogenic sources close to the measurement site. Levels of $O_3$ were very variable (20-80 ppb) with the highest levels encountered during the last part of the campaign. This period corresponded to the warmest period with intense biogenic emissions but also to air masses originating from the north of Italy, an area characterized by intense anthropogenic emissions of ozone precursors.

Oxygenated VOCs (including primary and secondary OVOCs from anthropogenic and biogenic origins) largely dominate the speciation of the measured VOCs (78%-80%, see Fig. S5 in supplements). OVOCs are dominated by methanol, acetone and formic acid which represent 28%, 23% and 14% of total OVOCs respectively. The weak contribution of biogenic hydrocarbons to the total VOC composition (4-5%, see Fig. S5 in supplements) is due to the fact that these contributions are calculated on a 24-h basis and not only during daytime when their concentrations are more elevated.

Finally, Anthropogenic NMHCs represent only 15-18% of the measured VOCs (see Fig. S5 in supplements), which is consistent with the remote location of the site. This VOC family is dominated by ethane, propane and ethylene which represent 34%, 7% and 7% of total A-NMHCs respectively. However, it is worth noting that this apportionment is only valuable for the measured species. Indeed, the difference between measured OH reactivity (total sink of OH) and calculated one, using all measured compounds, reported for this campaign indicates that approximately 56±15% (1σ, on average) of the measured OH reactivity was missing. The largest fraction of missing OH reactivity was observed between 23/07 and 30/07, a period associated to the Marine-West and South clusters (Zannoni et al., 2016). Therefore, a large fraction of the VOCs composing the air masses reaching the site has not been measured yet.

### 4.1.2. Aerosol phase

The chemical composition derived from Q-ACSM measurements is reported in Figure 5a for the period of study (15/07-05/08) and shows a clear and permanent dominance of OM which represents 55% of the total mass of NR-PM$_1$ on average (average of 3.74±1.80 µg m$^{-3}$), followed by sulphate (27%, 1.83±1.06 µg m$^{-3}$), ammonium (13%, 0.90±0.55 µg m$^{-3}$), and





nitrate (5%, $0.31\pm0.18$ µg m$^{-3}$). These values are in the range of the monthly mean
concentrations for summer calculated with Q-ACSM data over the two years period
measurements (June 2012-July 2014) performed at the measurement site (J. Sciare,
unpublished data). OM concentrations are comparable to those observed by Sciare et al.
(2008) in the Eastern Mediterranean for the month of July ([OC] = $2.18\pm0.65$ µg m$^{-3}$ and
using an OM-to-OC ratio of 1.9; Sciare et al., 2003). The overall OA concentrations during
the campaign vary within two orders of magnitude (ranging from 0.13 to 9.77 µg m$^{-3}$) with
very short periods (one to four hours) characterized by very sharp drops (close to zero)
associated to clouds passing at the station and subsequent uptake of fine aerosols into the
cloud droplets.

11          The temporal variability of OC and WSOC are reported in Figure 5b and show very

close patterns with, however, few periods with noticeable discrepancies (17/07; 28/07-30/07).
There is a clear correlation between the two datasets ($r^2$=0.68; N=229) with slope of 0.58
reflecting that more than half of OC is water-soluble. The correlation between OC (OCEC
Sunset Field Instrument) and OM (Q-ACSM) shows a better agreement ($r^2$=0.86; N=229)
with a slope of 0.87 when using an OC-to-OM ratio of 1.9. This slope close to one reflects the
general good agreement between both instruments measuring OC in PM$_{2.5}$ and OM in PM$_{1}$,
respectively. A closer look at the OM/OC ratio derived from these two instruments (not
shown) shows a slight but systematic diurnal variability with minimum values around 09:00
LT and a constant rise in the course of the day with a maximum value at 21:00 LT.
Interestingly, although the absolute OM/OC ratio calculated empirically from Q-ACSM mass
spectra (Aiken et al., 2008) should be interpreted with caution (Crenn et al., 2015), its
temporal variability shows exactly the same diurnal pattern of local photochemical oxidation
of OA, thus providing further consistency of our Q-ACSM fragmentation data which will be
used later in the source apportionment.

26          Real-time observations of two light organic tracers (MSA and oxalate) are reported in

Figure 5c. MSA (methanesulfonic acid, CH$_3$SO$_3$H) is an oxidation end-product of
dimethysulfide (DMS), a natural gas produced from the marine phytoplankton activity. MSA
is mostly in the aerosol phase and formed through the heterogeneous oxidation of
dimethysulfoxide (DMSO). It has been recently used to infer a marine organic aerosol (MOA)
source from a source apportionment study performed in the region of Paris (France) (Crippa
et al., 2013b). Oxalic acid is the most abundant dicarboxylic acid in the troposphere
(Kawamura et al., 1996). Its primary sources cannot solely explain its observed ambient
concentrations (Huang and Yu, 2007), suggesting that secondary formation processes remain





significant (Warneck, 2003). Simulations of these compounds predict reactions through in-
cloud processing (Carlton et al., 2007; Ervens et al., 2004, 2008; Fu et al., 2008; Lim et al.,
2005; Myriokefalitakis et al., 2011; Sorooshian et al., 2006; Volkamer et al., 2007; Warneck,
2003). Field measurements also brought evidence of heterogeneous chemistry in the
formation of oxalic acid through different routes (Crahan et al., 2004; Sorooshian et al., 2006,
2007). Consequently real-time observations of MSA and oxalate may be used here in our
source apportionment study to infer secondary oxidation processes.

## 4.2 Exploring the drivers of VOC variability at Cape Corsica

Source-receptor models, such as PMF, usually aim at identifying and quantifying the
contributions of sources of pollutants impacting a measurement site. In our case, the remote
location of the site combined with the reactivity of the selected species does not allow a
proper identification and quantification of primary sources. Our main objective, here, lies
within the identification of co-variation factors of species which could be representative of
aged or fresh primary emission but also of photochemical processes occurring during long
range transport or occurring locally. For this purpose, PMF was applied to a large dataset (42
different species) including primary VOCs from anthropogenic or biogenic origins but also
secondary products measured by three different techniques (PTR-ToF-MS, GC/FID-FID and
GC/FID-MS, see section 2.2).
Figure 6 shows the time series of the 6 factors obtained by the PMF analysis. Figure 7
shows the contributions of each factors to the species selected as inputs for the PMF model (in
%) as well as the absolute averaged contribution of each species to the 6 factors determined
by the PMF analysis (in ppt). Finally, Figure 8 presents the maps of simulated contributions
(in ppt) using the CF model for 4 of the 6 PMF factors. The relative contributions of the
different PMF factors to the sum of species used as inputs are presented in the Supplementary
Material as Fig. S6.

### 4.2.1 Anthropogenic influence

Among the 6 PMF factors, three different factors were attributed to primary
anthropogenic sources (Factors 2, 3, and 5) and are characterized by compounds of various
lifetimes (Figure 6 & Figure 7). The lifetimes reported below are estimated from kinetic rate





constants of the reactions between the species of interest and OH, assuming an averaged OH
concentration of $2.0\times10^{6}$ molecules $cm^{-3}$.
Factor 2 is composed of long-lived primary anthropogenic species such as ethane
(58% explained by Factor 2), acetylene (44% explained), propane (30% explained) and
benzene (45% explained) (see Figure 7) with lifetimes ranging from 5 to 25 days and typically
emitted by natural gas use and combustion processes. In addition to these long-lived primary
anthropogenic species, other anthropogenic NMHCs, with shorter lifetimes, compose this
factor, such as ethylene (35% explained) or 2-methyl-2-butene co-eluted with 1-pentene (42%
explained). It tends to indicate that in addition of the lifetime, the nature of the sources (e.g.
combustion processes) also partly influences the profile of this factor. Furthermore, Factor 2
exhibits a behaviour similar to CO (see supplementary material S7), a long-lived compound
(lifetime of ~24 days) mainly emitted by combustion processes, supporting the identification
of this factor as long-lived anthropogenic. Hence, the lack of diurnal variability in this factor
(see supplementary material S8) confirmed its long-range origin. The potential source areas
associated with this factor (Figure 8) are the North of Italy (Po Valley) and the South East of
France as well as, to a lower extent, the North-east of Tunisia (area of Tunis). These areas,
and particularly the Po valley, are known to supply high anthropogenic emissions due to
intense industrial activities and a dense road network (Thunis et al., 2009). This result
strengthens the assumption of primary anthropogenic origin for this factor. This factor
represents 16-17% of the sum of VOC species used as inputs in the PMF model
(supplementary material Fig. S6).
Factor 3 is composed by medium-lived primary anthropogenic species such as n-
pentane (78% explained by factor 3), iso-pentane (68% explained), 2,2dimethylbutane (48%
explained) (see Figure 7) with lifetimes ranging from 1 to 3 days and typically emitted by
gasoline evaporation or vehicle exhaust. This factor shows higher levels for air masses
coming from the Europe-North-East and the France-North-West sectors (see Figure 6).
Consequently, north of Italy (Po valley) and south-east of France, areas experiencing high
anthropogenic emissions, are also identified as potential source areas for this factor (Figure 8).
Potential source areas identified in the centre of France are most likely falsely attributed to
this area due to corridor effect: the air masses reaching Cape Corsica and passing over the
centre of France encompass systematically source areas (south-east of France). This factor
represents 12% of the sum of VOC species used as inputs in PMF model (supplementary
material Fig. S6).



Factor 5 is composed by short-lived primary anthropogenic VOCs such as ethylene
(38% explained by factor 5), propene (44% explained) and toluene (38% explained) with
lifetimes ranging from 5 to 23 h and typically emitted by combustion processes. This factor
exhibits higher levels for air masses coming from the Corsica-South sector (see Figure 6).
Likewise, areas at the south of Corsica are identified as potential source areas for this factor
(Figure 8). Emissions of these areas could be due to both intense ship emissions, which
speciation is dominated by alkenes (ethene, propene), aromatics and heavy alkanes ($>C_6$)
(Eyring et al., 2005). A contribution of the Corsican cities in this southern sector cannot be
excluded. This factor represents 21-23% of the sum of species used as inputs in the PMF
model (supplementary material Fig. S6).
The total contribution of anthropogenic-like factors to the sum of species used as
inputs of the PMF model is in the range 49-52%. This is higher than the contributions of
anthropogenic NMHCs relatively to measured VOCs (15%, see Figure S5 in supplement).
This can be explained by the fact that not only anthropogenic NMHCs contribute to these
anthropogenic factors and some OVOCs are also part of them. For example, methanol and
acetone contribute both in a non-negligible extent to these anthropogenic factors. Indeed,
methanol contributes to 7% and 39% of LL-Anthropogenic factor and ML-Anthropogenic
factor respectively; and acetone contributes to 14% and 11% of LL-Anthropogenic factor and
SL-Anthropogenic factor respectively. Therefore, higher contributions of these factors to the
gas phase composition are expected. Considering the primary anthropogenic part of OVOCs,
determined based on the anthropogenic factor contribution to OVOCs, the contribution of
anthropogenic VOCs to measured VOCs rises to 42% (see Figure 9), much closer to the PMF
results.

**4.2.2 Biogenic influence**

Among the 6 factors, two biogenic factors are also clearly identified (Factors 1 and 6).
They are composed respectively by primary biogenic species (Factor 1) and oxidation
products of primary biogenic hydrocarbons (Factor 6). Therefore, they have been classified
and will be reported in the following respectively as "primary biogenic factor" (Factor 1) and
"secondary biogenic factor" (Factor 6).
Indeed, Factor 1 is composed by primary biogenic species with very short lifetimes,
emitted locally by the vegetation surrounding the measurement site, such as isoprene (68%
explained by Factor 1), the sum of monoterpenes (83% explained) or camphor co-eluted with


undecane (38% explained) (see Figure 7). This factor exhibits clear diurnal cycles (Figure 6
and supplementary material Fig. S8) and is correlated, as expected, with temperature (see
supplementary material S7), which is well-known to influence biogenic emissions (Guenther
et al., 1995; 2000).

5           This factor represents 14% of the sum of species used as inputs in the PMF model

(supplementary material Fig. S6). This is higher than the contributions of biogenic NMHCs to
measured VOCs (4-5%, see Fig. S5 in supplement). As already proposed for anthropogenic
factors, this can be explained by the fact that not only biogenic NMHCs contribute to these
primary biogenic factors but some biogenic OVOCs can also be part of it. For example,
carboxylic acids, methanol and acetone also contribute (13%, 15% and 11% on average,
respectively explained by factor 1). Taking into account the primary biogenic part of OVOCs,
the contribution of biogenic VOCs to measured VOCs rises to 15% (see Figure 9), which is
closer to the PMF results.

14          Factor 6 is composed by oxidation products of primary biogenic VOCs such as

Methyl-Vinyl-Ketone (MVK) and methacrolein (MACR) (67% explained by Factor 6), which
are measured as a sum by PTR-ToFMS (m/z 71.05), nopinone (45% explained), and
pinonaldehyde (39% explained) (see Figure 7). More specifically, MVK and MACR are first-
generation oxidation products of isoprene (Miyoshi et al., 1994), nopinone is a first-
generation oxidation product of β-pinene (Wisthaler et al., 2001), and pinonaldehyde is a first-
generation oxidation product of α-pinene (Wisthaler et al., 2001). As expected, the variability
of this factor is also correlated to the temperature (see supplementary material S7). It can be
explained by higher emissions of primary biogenic VOCs under warmer conditions associated
with more intense photochemistry. Furthermore, the lowest levels of Factor 6 correspond to
the highest wind speed observed at the measurement site and vice-versa (see Figure 6); near
zero contributions of Factor 6 are observed on 23, 24 and 25 July when wind speeds were
comprised between 3 and 10 m s$^{-1}$. In contrast, the highest diurnal maxima were observed on
26, 27 and 28 July and on 02 and 03 August when wind speeds did not exceed 3 m s$^{-1}$. This
factor is characterized by first-generation oxidation products of primary biogenic VOCs
emitted in the vicinity of the site, and low wind speeds are necessary to observe them at
significant levels. In case of high wind speed, these oxidation products undergo fast transport
and dilution and low levels might be observed. This factor represents 6% of the sum of
species used as inputs in the PMF model (supplementary material Fig. S6) and is therefore the
less important.



### 4.2.3 Oxygenated Factor

The last factor (Factor 4) has been interpreted as "oxygenated factor" since it is mainly characterized by OVOCs such as carboxylic acids (54% formic acid, 43% acetic acid, 28% propionic acid, and 14% butyric acid), alcohols (49% methanol and 21% isopropanol), carbonyls (57% acetone, 18% acetaldehyde, and 21% methyl-ethyl-ketone). Most of these species are formed by the oxidation of both anthropogenic and biogenic compounds although some of them can also be directly emitted in the atmosphere and therefore can be of both primary and secondary origins. For example, methanol (the highest contributor to Factor 4) can be emitted by vegetation (MacDonald and Fall, 1993), biomass burning (Holzinger et al., 1999) or urban and industrial activities (Hu et al., 2011). It can also be formed by photochemistry (mainly photo-oxidation of methane) (Tyndall et al., 2001). The same stands for acetone (the 2[nd] highest contributor to Factor 4). Indeed, acetone can be directly emitted from vegetation (Goldstein and Schade, 2000; Hu et al., 2013), biomass burning (Simpson et al., 2011), and anthropogenic sources (Hu et al., 2013), and it can also be formed via photochemical oxidation of anthropogenic VOCs such as alkanes (Goldstein and Schade, 2000) and biogenic VOCs such as monoterpenes (Reissell et al., 1999). Note that the same stands for carboxylic acids which also have multiple sources (de Angelis et al., 2012 and references therein).

The multi-sources pattern for this factor is highlighted by its time series. Indeed, Factor 4 exhibits a similar behaviour than anthropogenic factors (Factors 2 and 3) at the beginning of the campaign with an increase to reach a maximum around the 21 July and then a decrease. This Factor rises again during the intense biogenic influenced warm period (26, 27 and 28 July) as observed for the secondary biogenic factor (Factor 6).

The CF analysis for this factor leads to the identification of the north of Italy and a large area in the southern of Corsica as potential source regions. North of Italy may contribute to the anthropogenic/continental influence of this factor while the large regions at the south of Corsica may contribute to the biogenic influence since the highest biogenic signature also corresponds to air masses coming from the Corsica-South sector. This could be explained by both potential biogenic emissions from vegetation of Corsica (the site being at the extreme north of the island) and/or warmer and more stagnant conditions arising when air masses came from Corsica-South sector favouring local biogenic emissions and low dispersion of oxidation products. It can also be due to local anthropogenic emissions from Corsica cities



erroneously attributed to more distant regions as already observed for the CF analysis of
Factor 5. Finally, one cannot rule out the possibility of a primary or secondary influence of
ship emissions to Factor 4 for this potential source area. This is also in accordance with the
non-negligible contribution of this factor to the acetylene variability (29% explained by this
factor). This factor represents 28-31% of the sum of species used as inputs in the PMF model
(supplementary material Fig. S6) and is therefore the most important one. Combined with the
secondary biogenic factor it leads to a contribution of 34-37% for the oxygenated factors. This
is significantly lower than the OVOC contribution to the actual measured VOCs (80%, see
Fig. S5) and can be explained by the contribution of most OVOCs such as acetone, methanol
or carboxylic acids to other PMF factors. Only considering the secondary part of measured
OVOCs, their contribution to measured VOCs decreases to 42% (see Figure 9), which is
closer to the PMF results.

### 4.2.4 Apportionment of measured OVOC

From the 6 PMF factors, it is possible to apportion the measured OVOCs among their
potential different origins (primary anthropogenic or biogenic emissions, photochemical
production from the oxidation of anthropogenic or biogenic hydrocarbons). Therefore, Factor
1 is attributed to a primary biogenic origin, Factors 2, 3 and 5 are attributed to a primary
anthropogenic origin and factors 4 and 6 are attributed to a secondary origin (photochemical
oxidation of primary VOCs from both biogenic and anthropogenic origins). To do so, the
contributions of each OVOC to a specific PMF factor are summed up and ascribed to the
corresponding origin. Subtracted backgrounds of acetone and methanol are redistributed to
each PMF factors accordingly to the relative contribution of these species to each factor. The
apportionment of anthropogenic, biogenic and secondary origin for OVOCs can be seen in
Figure 9. Primary anthropogenic sources, primary biogenic sources and secondary processes
account for 34%, 13% and 53% of the measured OVOCs, respectively. Therefore, measured
OVOCs at cape Corsica are approximately half oxidation products of VOCs and half primary
VOCs.

### 4.2.5 Comparison with other PMF studies performed in remote environments

To our best knowledge, only three studies have been conducted applying PMF for gas
phase species in remote environments (Sauvage et al., 2009; Lanz et al., 2009; Leuchner et al.,



2015). Moreover, these studies were only based on NMHC measurements, and chlorinated
organic species for one of them (Lanz et al., 2009). No oxygenated VOCs were considered.
Consequently, these three studies only identified factors representative of primary sources.
Indeed, Leuchner et al. (2015) identified 6 PMF factors, at a remote site at
Hohenpeissenberg over a period of 7 years (2003-2009), including primary biogenic, short-
lived combustion, short-lived evaporative, residential heating, long-lived evaporative and
background factors. Therefore, the classification of factors was linked to the difference in the
sources typology (biogenic vs anthropogenic, combustion vs evaporative) and/or the lifetime
of compounds (short-lived vs long-lived). Lanz et al. (2009) found only 4 PMF factors, at a
continental mountain site at Jungfraujoch (Switzerland) during height years (2000-2007),
including a highly aged combustive emissions factor correlated to CO, a fresh emissions and
solvent-use factor correlated to $NO_x$, and two industrial factors mainly explaining the
variability of chlorinated compounds. Sauvage et al. (2009) applied PMF to a database of
NMHCs measured at three background sites in France, leading to 5 common PMF factors
including evaporative sources, residential heating, vehicle exhaust, remote sources attributed
to aged background air and biogenic emissions.
Therefore, we incorporated for the first time OVOCs in a database used for PMF
analysis at a remote environment. It allows the first identification of PMF factors
representatives of secondary processes in addition to factors related to primary sources. As it
has been found in previous studies performed in such environments, we also found that
primary anthropogenic PMF factors were separated according to the lifetime of compounds
which composed them. As in the three studies described above, a clear primary biogenic
factor is identified in our study. Furthermore, our analysis allowed the apportionment of the
anthropogenic, biogenic and secondary parts of OVOCs.
**4.3 Source apportionment of OA at cape Corsica**
Based on the two available months of ACSM data, a 3-factor solution was selected
here, corresponding to a minimum of the quality parameter $Q/Q_{exp}$. Mass spectra reported in
Figure 10 show a typical HOA (Hydrogen-like OA) profile for the first factor with n-alkanes,
branched alkanes, cycloalkanes, and aromatics, leading to high signals at the ion series
$C_nH^+_{2n+1}$ and $C_nH^+_{2n-1}$ (m/z 27, 29, 41, 43, 55, 57, 69, 71, etc.) and typical for fossil fuel
combustion (Canagaratna et al., 2004; Chirico et al., 2010). We have also used the terms "SV-
OOA" (Semi-Volatile-Oxygenated Organic Aerosol) and "LV-OOA" (Low-Volatile OOA) as





introduced by Jimenez et al. (2009) to describe the two remaining factors. In these two
factors, m/z 43 and m/z 44 are the most prominent peaks which is consistent with OOA
(Oxygenated OA) spectra and the m/z 44 to m/z 43 ratio that increases with aging (Ng et al.,
2010a). The signal at m/z 43 is dominant for the second factor and mainly comes from the
fragmentation of either hydrocarbon chains to form $C_3H_7^+$ or carbonyls to form $C_2H_3O^+$;
therefore this factor appears to be the less oxidized and was named consequently SV-OOA.

7       The consistency of the different OA factors was further checked with external tracers

in Figure 11; HOA with BC (fossil fuel tracer), SV-OOA with WSOC, and LV-OOA with
oxalate. The good agreement of SV-OOA with WSOC is consistent with freshly formed SOA
being semi-volatile and water-soluble as reported for instance by Hennigan et al. (2008a) who
observed strong similarities between semi-volatile $NH_4NO_3$ and (PILS-TOC based) WSOC.
The good agreement between LV-OOA with oxalate is consistent with the fact that both are
mostly composed by carboxylic acid COO chains and the use of oxalate as a proxy of highly
oxidized OA as stated before. Note also that good correlation is obtained between the
averaged OOA mass spectra taken from Ng et al. (2010b) and our two factors with correlation
coefficients ($r^2$) of 0.96 and 0.81 for our SV-OOA and LV-OOA factors, respectively.

17       The different OA factors obtained here are mainly of continental origin and therefore,

their temporal variability is mostly related to the amount and frequency of continental air
masses reaching the sampling site. Nevertheless, the diurnal variation of SV-OOA and LV-
OOA (Fig S9) suggest that local photochemical processes have also occurred, with local
formation of fresh SV-OOA in the morning followed by a rapid oxidation which could
explain the enhancement of LV-OAA in the afternoon.

23       Average mass concentrations are 0.13, 1.59, and 1.92 µg m$^{-3}$ for the 3 determined

factors HOA, SV-OOA, and LV-OOA, respectively, for a total average OA concentration of
3.63 µg m$^{-3}$ and a contribution of OA to NR-PM$_1$ of 52%. As a result, secondary OA
represent about 96% of OA with aged (LV-)OOA contributing for approximately 55% of this
secondary OA fraction. In the recent years, increasing background OA observations have
become available in the Mediterranean, mostly at coastal sites located in the Northern part of
the basin (Spain, France, Italy, and Greece). For instance, long-term (13-months) ACSM
measurements were performed at a regional background site in the western Mediterranean
(Spain), located in the Montseny natural park 50 km North-North-East of Barcelona,
approximately 500 km west of Cape Corsica, and reported observations similar to ours with
an OA contribution to NR-PM$_1$ of c.a. 60% (Minguillon et al., 2015), three major OA sources
(HOA, SV-OOA, and LV-OOA) during summer with again a very prominent secondary
fraction (85% of OA), and OA profiles very similar to those obtained here.
**4.4 Gas-aerosol link**
First, the gas-phase "Oxygenated Factor" (Factor 4) is correlated to the organic
fraction of the aerosol measured by ACSM ($R^2$=0.58, n=498; see supplementary material S7).
This fair correlation likely highlights the close link between gaseous oxidation products
observed at the site and measured Organic Aerosol (OA) since they stem from similar
processes. During the campaign, very low levels of Primary Organic Aerosols were observed
(HOA determined by ACSM measurements below 0.4 µg m$^{-3}$, see Figure 11a (top panel)).
Thus, this correlation is most likely due to the secondary fraction of OA, representing 96% of
OA (see section 4.3), which can come from the oxidation of both biogenic and anthropogenic
gaseous precursors, explaining the similar behaviour as Factor 4.
Figure 12 shows stacked time series of the different fractions (inorganic and organic)
of aerosol measured by ACSM (top panel) as well as stacked time series of contributions of
PMF factors (middle panel) for the VOCs (see section 4.2). This figure aims at drawing a
parallel between aerosol and gas phase compositions to highlight the link between the two
phases.
From these graphs and from the back-trajectory clusters (also shown in Figure 12), it
is possible to distinguish two periods during which processed anthropogenic/continental air
masses reached the site (between 19 and 27 July and between 30 July and 03 August 2013).
The first period is characterized by high contributions of anthropogenic and oxygenated gas
PMF factors (middle panel of Figure 12) as well as an aerosol with inorganic (ammonium
sulphate) and organic fractions in approximately similar proportions (top panel of Figure 12).
This period also corresponds to the highest values of ln(propane/ethane) (-1.4 on average and
up to -0.8, see bottom panel of Figure 12) hence the less aged air masses, coinciding with the
Europe-North-East    sector.   The   second   period   of   long-range   transported
anthropogenic/continental emissions is characterized by less intense anthropogenic gas phase
PMF factors, especially for the long-lived anthropogenic factor, and a clear predominance of
the organic fraction for aerosols. Aerosol mass concentrations are also lower by
approximately 50% compared to the first period. During both periods, a non-negligible
biogenic influence is also observed from primary and secondary biogenic PMF VOC factors.
This is even more pronounced for the second "anthropogenic" period. During these periods it



is, therefore, likely that oxygenated VOCs and OOA have both biogenic and anthropogenic
origins in variable proportions.
A period of intense biogenic influence, without significant long-range transport of
anthropogenic/continental emissions, can also be distinguished (between 26 and 28 July) with
elevated contributions of the primary and secondary biogenic gas-phase PMF factors (Figure
12). The oxygenated gas-phase PMF factor also rose during this period and the aerosol
composition is dominated by OA with low levels of inorganic aerosols. Indeed, the inorganic
fraction of aerosols decreases to reach less than 10% of the aerosol composition on 27 July.
This strong decrease occurred at the same time than a change of air mass origin from Marine-
West to Corsica-South. This is consistent with the lack of anthropogenic influence during this
period, confirmed by lower ln(propane/ethane) (-1.8 on average up to -2.3, see bottom panel
of Figure 12). It is, therefore, likely that the oxygenated VOCs and the organic fraction of
aerosols during these days are mainly influenced by biogenic sources.
Finally, very low contributions of HOA were observed during the whole campaign
from the PMF analysis of ACSM measurements (typically below 0.3 µg m$^{-3}$ all along the
campaign). This illustrates the weak influence of freshly emitted primary anthropogenic
sources of OA at the site. This is also confirmed by low levels of black carbon (BC<0.9 µg m$^{-}$
$^{3}$ for the whole campaign, see Figure 11a).
An analysis of the isotopic ratio of $^{14}$C in aerosol sampled at cape Corsica reveals that
the organic fraction of the aerosol measured during the ChArMEx SOP2 field campaign
mainly came from biogenic sources and the oxidation of biogenic VOCs with measured non-
fossil OC of 2.42±0.86 µgC m$^{-3}$ on average (76±3 % of OC on average). The secondary and
primary anthropogenic sources to OC represented by measured fossil OC was
0.44±0.22 µgC m$^{-3}$ on average with a contribution to OC being 14±3 % of OC on average.
Elementary carbon contributed to only 10% of total carbon during the campaign with
averaged measure biomass EC and fossil EC being 0.16±0.06 µgC m$^{-3}$ and 0.17±0.06 µgC m$^{-}$
$^{3}$, respectively. Results from this analysis will be presented in more details in a forthcoming
paper (Pey et al., in preparation).
Given the good correlation observed between OA and the gas-phase oxygenated factor
(R$^{2}$=0.58, n=498), a common origin can be attributed to both OA and OVOCs observed at
Cape Corsica. Therefore, a predominance of secondary biogenic origin, during the whole
campaign, is likely for OVOCs, such as acetone, methanol or carboxylic acids for example,
which composed the oxygenated PMF factor. As stated previously, this is also consistent with
the large fraction of WSOC in OA, whose fraction usually refers to biogenic SOA.



Nevertheless, a less important but still significant secondary anthropogenic origin is also
likely for OVOCs.
**5 Conclusions**
The ChArMEx SOP2 field campaign provided a unique opportunity to give insights in
the various sources and fates of organic carbon in the Mediterranean atmosphere, thanks to
the measurement of a large panel of gaseous and aerosol species at a remote site located at
Cape Corsica in the western Mediterranean basin. The combination of gaseous and particulate
organic databases, as collected during this campaign, is not common and has the potential to
help improving our understanding of SOA formation. Moreover, the Mediterranean basin is
an ideal location to characterize organics in the atmosphere since it is impacted by strong
natural and anthropogenic sources and undergoes intense photochemical ageing especially
during summer. The measurement site (Cape Corsica) offered ideal experimental conditions
since it is surrounded by the sea and it is located at various distances from regional
anthropogenic emissions hotspots (such as north of Italy, south-east of France, north-east of
Spain or north of Africa). These characteristics coupled to extremely low local anthropogenic
sources allowed the study of anthropogenic plumes after several days of atmospheric
processing. In addition, intense local biogenic emissions permitted the investigation of
biogenic and anthropogenic interactions on air mass composition.
These specific conditions led to the observation of contrasted situations, i.e. highly
variable photochemical ages of processed anthropogenic air masses coupled with intense and
local biogenic emissions. Low levels of anthropogenic VOCs (<250 ppt for acetylene for
example) were overall observed, confirming the remoteness of the site. In contrast, significant
levels of short-lived biogenic VOCs (up to 1.2 and 2.0 ppb for isoprene and the sum of
monoterpenes, respectively) were observed. Elevated mixing ratios of OVOCs (e.g. up to 3.8
ppb for acetone) were also measured during the campaign due to the oxidation of both
biogenic and anthropogenic precursors. These OVOCs exhibit the largest contribution to the
VOC budget.
The aerosol chemical composition derived from Q-ACSM measurements shows a
clear predominance of OM, which represents 55% of the total mass of $NR-PM_1$ on average,
followed by sulphate (27%), ammonium (13%), and nitrate (5%). Furthermore, the temporal
variability of OC and WSOC shows very similar patterns, leading to a clear linear correlation



between the two datasets (r²=0.68). The slope found is 0.58, highlighting that more than half
of OC is water-soluble.

3        PMF was conducted to identify co-variation factors of VOCs that are representative of

primary emissions as well as secondary photochemical transformations occurring during the
transport of air masses. This analysis was performed using a gas-phase database of 42 VOCs
(or sum of VOCs) of anthropogenic and biogenic origins, including NMHCs and for the first
time OVOCs. A 6-factor solution turned out to be optimum for this PMF analysis. In parallel,
a concentration field (CF) analysis was conducted on 4 PMF factor to help in their
identification through the localization of potential source areas. This combination of CF and
PMF was particularly helpful to interpret factors associated to long-range transport of
anthropogenic compounds.

12        Three anthropogenic factors characterized by primary anthropogenic VOCs of various

lifetimes were found. The CF analysis confirmed the anthropogenic nature of these factors by
an identification of potential source areas in region experiencing intense anthropogenic
activities (e.g. Po valley and south-east of France).

16        Two biogenic factors were also identified. Both factors exhibited clear diurnal cycles

and were correlated to temperature. In addition to a primary biogenic factor, usually observed
in VOC source apportionment studies, we also clearly identified, for the first time in PMF
analysis, a secondary biogenic factor made of first-generation oxidation products of biogenic
VOCs.

21        A last oxygenated factor characterized by OVOCs of both biogenic and anthropogenic

origins, was also derived from the PMF analysis. The identification of this unusual factor was
made possible by the extension of the input database to secondary oxygenated VOCs. This
factor was influenced by anthropogenic and biogenic sources showing elevated levels during
both periods of intense local biogenic influence (e.g. 26-28 July) and periods of long-range
transport of anthropogenic/continental emissions (e.g. 21-23 July). This factor was also
correlated to submicron OA measured by ACSM ($R^2$=0.58, n=498), highlighting the close
link between secondary OVOCs and (secondary) OA at cape Corsica. The CF analysis of this
factor suggested potential source areas that could be attributed to both
anthropogenic/continental (North of Italy) and biogenic influences (areas at the south of
Corsica).

32        The source apportionment of OA measured by ACSM led to a 3-factor solution

identified as hydrogen-like OA, semi-volatile oxygenated OA, and low-volatile oxygenated
OA. These 3 factors accounted for an averaged mass concentration of 0.13, 1.59, and 1.92 µg



m$^{-3}$, respectively, for a total OA mass concentration of 3.63 µg m$^{-3}$, mainly associated to
secondary formation (96%).
A coupled analysis of VOC and OA sources was conducted. During biogenic periods,
the aerosol composition was dominated by (secondary) OA, while during periods of long-
range transport of anthropogenic/continental emissions, the inorganic and organic fractions of
submicron aerosol were similar. During the whole campaign, low levels of Hydrogen-like OA
(HOA) were observed (<0.3 µg m$^{-3}$), indicating a weak influence of primary anthropogenic
sources on OA. Finally, the analysis of the isotopic ratio of $^{14}$C in PM$_1$, which will be
presented in detail in a forthcoming paper, highlighted that OA aerosols were mainly
produced from the local oxidation of biogenic VOCs. Given the good correlation between the
oxygenated gas phase PMF factor and organic aerosols, the same origin is likely for OVOCs
with a predominance of sources from the oxidation of BVOCs.

**Acknowledgements**

This study received financial support from Mistrals / ChArMEx programmes,
ADEME, the French environmental ministry, the CaPPA projects and the Communauté
Territoriale de Corse (CORSiCA project). The CaPPA project (Chemical and Physical
Properties of the Atmosphere) is funded by the French National Research Agency (ANR)
through the PIA (Programme d'Investissement d'Avenir) under contract "ANR-11-LABX-
0005-01" and by the Regional Council Nord-Pas de Calais and the "European Funds for
Regional Economic Development" (FEDER). This research was also funded by the European
Union Seventh Framework Programme under Grant Agreement number 293897, "DEFIVOC"
project, CARBOSOR/Primequal and SAF-MED (ANR, grant number: ANR-12-BS06-0013-
02). Greenhouse gases data were provided by the ICOS-France monitoring network.
The authors also want to thank Eric Hamonou and François Dulac for logistic help
during the campaign as well as all the participants of the ChArMEx SOP2 field campaign.





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

Reproducibility of concentration and fragment results from 13 individual Quadrupole Aerosol
Chemical Speciation Monitors (Q-ACSM) and consistency with co-located instruments,
Atmos. Meas. Tech., 8, 5063-5087, doi:10.5194/amt-8-5063-2015, 2015.
Crippa, M., Canonaco, F., Slowik, J. G., El Haddad, I., DeCarlo, P. F., Mohr, C., Heringa, M.
F., Chirico, R., Marchand, N., Temime-Roussel, B., Abidi, E., Poulain, L., Wiedensohler, A.,
Baltensperger, U., and Prévôt, A. S. H.: Primary and secondary organic aerosol origin by
combined gas-particle phase source apportionment, Atmos. Chem. Phys., 13, 8411–8426,
doi:10.5194/acp-13-8411-2013, 2013a
Crippa, M., El Haddad, I., Slowik, J. G., DeCarlo, P. F., Mohr, C., Heringa, M. F., Chirico,
R., Marchand, N., Sciare, J., Baltensperger, U., and Prévôt, A. S. H.: Identification of marine
and continental aerosol sources in Paris using high resolution aerosol mass spectrometry, J.
Geophys. Res.-Atmos., 118, 1950–1963, 2013b.





de Angelis, M., Traversi, R., Udisti, R.: Long-term trends of mono-carboxylic acids in
Antarctica: comparison of changes in sources and transport processes at the two EPICA deep
drilling sites, Tellus B, 64, 17331, DOI: 10.3402/tellusb.v64i0.17331, 2012
de Gouw, J. and Warneke, C.: Measurements of volatile organic compounds in the earth's
atmosphere using proton-transferreaction mass spectrometry, Mass. Spectrom. Rev., 26, 223–
257, doi:10.1002/mas.20119, 2007.
Detournay, A., Sauvage, S., Locoge, N., Gaudion, V., Leonardis, T., Fronval, I., Kaluzny, P.,
and Galloo, J.-C.: Development of a sampling method for the simultaneous monitoring of
straightchain alkanes, straight-chain saturated carbonyl compounds and monoterpenes in
remote areas. J. Environ. Monitor., 13, 983– 990, 2011.
Draxler, R. R. and Hess, G. D.: An overview of the HYSPLIT 4 modeling system for
trajectories, dispersion, and deposition, Aust. Meteorol. Mag., 47, 295–308, 1998.
Ervens, B., Feingold, G., Frost, G. J., and Kreidenweis, S. M., A modeling study of aqueous
production of dicarboxylic acids: 1. Chemical pathways and speciated organic mass
production, J. Geophys. Res.-Atmos., 109(D15), 2004
Ervens, B., Carlton, A. G., Turpin, B. J., Altieri, K. E., Kreidenweis, S. M., and Feingold, G.,
Secondary organic aerosol yields from cloud-processing of isoprene oxidation products,
Geophys. Res. Let., 35(2), 2008
Eyring, V., Kohler, H. W., van Aardenne, J., and Lauer, A.: Emissions from international
shipping: 1. The last 50 years, J. Geophys. Res., 110, D17305, doi:10.1029/2004JD005619,
21  2005.

Fares, S., Park, J. H., Gentner, D. R., Weber, R., Ormeno, E., Karlik, J., and Goldstein, A. H.:
Seasonal cycles of biogenic volatile organic compound fluxes and concentrations in a
California citrus orchard, Atmos. Chem. Phys., 12, 9865–9880, doi:10.5194/acp- 12-9865-
25  2012, 2012.

Forster, P., Ramaswamy, V., Artaxo, P., Berntsen, T., Betts, R., Fahey, D. W., Haywood, J.,
Lean, J., Lowe, D. C., Myhre, G., Nganga, J., Prinn, R., Raga, G., Schulz, M., and Van
Dorland, R.: Radiative Forcing of Climate Change, in: Climate Change 2007: The Physical
Science Basis, Contribution of Working Group I to the Fourth Assessment Report of the
Intergovernmental Panel on Climate Change, edited by: Solomon, S., Qin, D., Manning, M.,
Chen, Z., Marquis, M., Averyt, K. B., Tignor, M., and Miller, H. L., 2007
Fröhlich, R., Crenn, V., Setyan, A., Belis, C. A., Canonaco, F., Favez, O., Riffault, V.,
Slowik, J. G., Aas, W., Aijälä, M., Alastuey, A., Artiñano, B., Bonnaire, N., Bozzetti, C.,
Bressi, M., Carbone, C., Coz, E., Croteau, P. L., Cubison, M. J., Essergietl, J. K., Green, D.
C., Gros, V., Heikkinen, L., Herrmann, H., Jayne, J. T., Lunder, C. R., Minguillón, M. C.,
Mocnik, G., O'Dowd, C. D., Ovadnevaite, J., Petralia, E., Poulain, L., Priestman, M., Ripoll,
A., Sarda-Estève, R., Wiedensohler, A., Baltensperger, U., Sciare, J., and Prévôt, A. S. H.:
ACTRIS ACSM intercomparison – Part 2: Intercomparison of ME-2 organic source
apportionment results from 15 individual, co-located aerosol mass spectrometers, Atmos.
Meas. Tech., 8, 2555–2576, doi: 10.5194/amt-8-2555-2015, 2015.




Fu, T. M., Jacob, D. J., Wittrock, F., Burrows, J. P., Vrekoussis, M., and Henze, D. K., Global budgets of atmospheric glyoxal and methylglyoxal, and implications for formation of secondary organic aerosols, J. Geophys. Res.-Atmos., 113(D15), 2008

Gaimoz, C., Sauvage, S., Gros, V., Herrmann, F., Williams, J., Locoge, N., Perrussel, O., Bonsang, B., d'Argouges, O., SardaEsteve, R., and Sciare, J.: Volatile Organic Compounds sources in Paris in spring 2007, Part II: Source apportionment using positive matrix factorisation, Environ. Chem., 8, 91–103, doi:10.1071/en10067, 2011

Goldstein, A. H. and Galbally, I. E.: Known and unexplored organic constituents in the earth's atmosphere, Environ. Sci. Technol., 41, 1514–1521, 2007

Goldstein, A. H. and Schade, G. W.: Quantifying biogenic and anthropogenic contributions to acetone mixing ratios in a rural environment, Atmos. Environ., 34, 4997–5006, 2000.

Guenther, A., Hewitt, C. N., Erickson, D., Fall, R., Geron, C., Graedel, T., Harley, P., Klinger, L., Lerdau, M., McKay, W. A., Pierce, T., Scholes, B., Steinbrecher, R., Tallamraju, R., Taylor, J., and Zimmerman, P.: A global model of natural volatile organic compound emissions, J. Geophys. Res., 100, 8873–8892, 1995

Guenther, A., Geron, C., Pierce, T., Lamb, B., Harley, P., and Fall, R.: Natural emissions of non-methane volatile organic compounds, carbon monoxide, and oxides of nitrogen from North America, Atmos. Environ., 34, 2205–2230, 2000.

Heald, C. L., Jacob, D. J., Turquety, S., Hudman, R. C., Weber, R. J., Sullivan, A. P., Peltier, R. E., Atlas, E. L., de Gouw, J. A., Warneke, C., Holloway, J. S., Neuman, J. A., Flocke, F. M., and Seinfeld, J. H.: Concentrations and sources of organic carbon aerosols in the free troposphere over North America, J. Geophys. Res., 111, D23S47, doi: 10.1029/2006JD007705, 2006.

Hennigan, C. J., M. H. Bergin, J. E. Dibb, , and R. J. Weber, Enhanced secondary organic aerosol formation due to water uptake by fine particles, Geophys. Res. Lett., 35, L18801, doi:10.1029/2008GL035046, 2008a

Hennigan, C. J., Bergin, M. H., and Weber, R. J.: Correlations between water-soluble organic aerosol and water vapor: a synergistic effect from biogenic emissions?, Environ. Sci Technol., 42, 9079–9085, 2008b

Hildebrandt, L., Engelhart, G. J., Mohr, C., Kostenidou, E., Lanz, V. A., Bougiatioti, A., DeCarlo, P. F., Prevot, A. S. H., Baltensperger, U., Mihalopoulos, N., Donahue, N. M., and Pandis, S. N.: Aged organic aerosol in the Eastern Mediterranean: the Finokalia Aerosol Measurement Experiment – 2008, Atmos. Chem. Phys., 10, 4167–4186, doi:10.5194/acp-10-4167-2010, 2010.

Holzinger, R., Warneke, C., Hansel, A., Jordan, A., Lindinger, W., Scharffe, D.H., Schade, G., and Crutzen, P. J.: Biomass burning as a source of formaldehyde, acetaldehyde, methanol, acetone, acetonitrile, and hydrogen cyanide, Geophys. Res. Lett., 26, 1161–1164, doi:10.1029/1999GL900156, 1999.

Holzinger, R., Lee, A., Paw, K. T., and Goldstein, U. A. H.: Observations of oxidation products above a forest imply biogenic emissions of very reactive compounds, Atmos. Chem. Phys., 5, 67–75, doi:10.5194/acp-5-67-2005, 2005.



Hopke, P. K.: A Guide to Positive Matrix Factorization, EPA Workshop Proceedings, Materials from the Workshop on UNMIX and PMF as Applied to PM2.5, 14–16 February, 2000.

Hopke, P. K.: Recent developments in receptor modeling, J. Chemometr., 17, 255–265, doi:10.1002/cem.796, 2003

Hu, L., Millet, D. B., Mohr, M. J., Wells, K. C., Griffis, T. J., and Helmig, D.: Sources and seasonality of atmospheric methanol based on tall tower measurements in the US Upper Midwest, Atmos. Chem. Phys., 11, 11145–11156, doi:10.5194/acp-11- 11145-2011, 2011.

Hu, L., Millet, D. B., Kim, S. Y., Wells, K. C., Griffis, T. J., Fischer, E. V., Helmig, D., Hueber, J., and Curtis, A. J.: North American acetone sources determined from tall tower measurements and inverse modeling, Atmos. Chem. Phys., 13, 3379– 3392, doi:10.5194/acp-13-3379-2013, 2013.

Huang, X. F., and Yu, J. Z., Is vehicle exhaust a significant primary source of oxalic acid in ambient aerosols?, Geophys. res. Let., 34(2), 2007

Hwang, I. and Hopke, P. K.: Estimation of source apportionment and potential source locations of PM2.5 at a west coastal IMPROVE site, Atmos. Environ., 41, 506–518, 2007.

Jimenez, J. L., M. R. Canagaratna, N. M. Donahue, A. S. H. Prevot, Q. Zhang, J. H. Kroll, P. F. DeCarlo, J. D. Allan, H. Coe, N. L. Ng, A. C. Aiken, K. S. Docherty, I. M. Ulbrich, A. P. Grieshop, A. L. Robinson, J. Duplissy, J. D. Smith, K. R. Wilson, V. A. Lanz, C. Hueglin, Y. L. Sun, J. Tian, A. Laaksonen, T. Raatikainen, J. Rautiainen, P. Vaattovaara, M. Ehn, M. Kulmala, J. M. Tomlinson, D. R. Collins, M. J. Cubison, E., J. Dunlea, J. A. Huffman, T. B. Onasch, M. R. Alfarra, P. I. Williams, K. Bower, Y. Kondo, J. Schneider, F. Drewnick, S. Borrmann, S. Weimer, K. Demerjian, D. Salcedo, L. Cottrell, R. Griffin, A. Takami, T. Miyoshi, S. Hatakeyama, A. Shimono, J. Y Sun, Y. M. Zhang, K. Dzepina, J. R. Kimmel, D. Sueper, J. T. Jayne, S. C. Herndon, A. M. Trimborn, L. R. Williams, E. C. Wood, A. M. Middlebrook, C. E. Kolb, U. Baltensperger, and D. R. Worsnop, Evolution of Organic Aerosols in the Atmosphere, Science, 326, 1525-1529, 2009

Jobson, B. T., Wu, Z., Niki, H., and Barrie, L. A.: Seasonal trends of isoprene, C2-C5 alkanes, and acetylene at a remote boreal site in Canada, J. Geophys. Res., 99, 1589–1599, 1994

Kanakidou, M.A., Seinfeld, J.H., Pandis, S.N., Barnes, I., Dentener, F., Facchini, M.C., Van Dingenen, R., Ervens, B., Nenes, A., Nielsen, C.J., Swietlicki, E., Putaud, J.P., Balkanski, Y., Fuzzi, S., Horth, J., Moortgat, G., Winterhalter, R., Myhre, C.E., Tsigaridis, K., Vignati, E., Stephanou, E.G., Wilson, J., Organic aerosol and global climate modelling: a review, Atmos. Chem. and Phys., 5, 1053-1223, 2005

Kawamura, K., Kasukabe, H., and Barrie, L. A., Source and reaction pathways of dicarboxylic acids, ketoacids and dicarbonyls in arctic aerosols: One year of observations, Atmos. Environ., 30(10), 1709-1722, 1996

Kondo, Y., Miyazaki, Y., Takegawa, N., Miyakawa, T., Weber, R. J., Jimenez, J. L., Zhang, Q., and Worsnop, D. R.: Oxygenated and water-soluble organic aerosols in Tokyo, J. Geophys. Res., 112, D01203, doi: 10.1029/2006jd007056, 2007.



Kroll, J. H. and Seinfeld, J. H.: Chemistry of secondary organic aerosol: Formation and evolution of low-volatility organics in the atmosphere, Atmos. Environ., 42, 3593-3624, 2008

Lanz, V. A., Henne, S., Staehelin, J., Hueglin, C., Vollmer, M. K., Steinbacher, M., Buchmann, B., and Reimann, S.: Statistical analysis of anthropogenic non-methane VOC variability at a European background location (Jungfraujoch, Switzerland), Atmos. Chem. Phys., 9, 3445–3459, doi:10.5194/acp-9-3445-2009, 2009

Latella, A., Stani, G., Cobelli, L., Duane, M., Junninen, H., Astorga, C., and Larsen, B. R.: Semicontinuous GC analysis and receptor modelling for source apportionment of ozone precursor hydrocarbons in Bresso, Milan, 2003, J. Chrom. A, 39, 29–39, 2005

Lee, A., Goldstein, A. H., Keywood, M. D., Gao, S., Varutbangkul, V., Bahreini, R., Ng, N. L., Flagan, R. C., and Seinfeld, J. H.: Gas-phase products and secondary aerosol yields from the ozonolysis of ten different terpenes, J. Geophys. Res.-Atmos., 111, D07302, doi:10.1029/2005jd006437, 2006.

Lelieveld J., Berresheim H., Borrmann S., Crutzen P.J., Dentener F.J., Fischer J., Flatau P.J., Heland J., Holzinger R., Korrmann R., Lawrence M.G., Levi Z., Markowicz K.M., Mihalopoulos N., Minikin A., Ramanathan V., De Reus M., Roelofs G.J., Scheeren H.A., Sciare J., Schlager H., Schultz M., Seigmund, P., Steil B., Stephanou E.G., Steir P., Traub M., Warneke C., Williams J., Ziereis H., Global air pollution crossroads over the Mediterranean, Science, 298, 794-799, 2002

Leuchner, M. and Rappenglück, B.: VOC source–receptor relationships in Houston during TexAQS-II, Atmos. Environ., 44, 4056– 4067, doi:10.1016/j.atmosenv.2009.02.029, 2010

Leuchner, M., Gubo, S., Schunk, C., Wastl, C., Kirchner, M., Menzel, A., and Plass-Dülmer, C.: Can positive matrix factorization help to understand patterns of organic trace gases at the continental Global Atmosphere Watch site Hohenpeissenberg?, Atmos. Chem. Phys., 15, 1221-1236, doi:10.5194/acp-15-1221-2015, 2015.

Lim, H. J., Carlton, A. G., and Turpin, B. J.: Isoprene forms secondary organic aerosol through cloud processing: Model simulations, Environ. Sci. Technol., 39, 4441–4446, 2005.

Lohmann, U. and Feichter, J.: Global indirect aerosol effects: a review, Atmos. Chem. Phys., 5, 715–737, doi:10.5194/acp-5-715-2005, 2005

MacDonald, R.C. and Fall, R.: Detection of substantial emissions of methanol from plants to the atmosphere, Atmos. Environ., 27, 1709–1713, doi:10.1016/0960-1686(93)90233-O, 1993

Madronich, S.: Chemical evolution of gaseous air pollutants downwind of tropical megacities: Mexico City case study, Atmos. Environ., 40, 6012–6018, doi:10.1016/j.atmosenv.2005.08.047, 2006

McKeen, S. A., Liu, S. C., Hsie, E.-Y., Lin, X., Bradshaw, J. D., Smyth, S., Gregory, G. L., and Blake, D. R.: Hydrocarbon ratios during PEM-WEST A: a model perspective, J. Geophys. Res., 101, 2087–2109, doi:10.1029/95JD02733, 1996.

Minguillón, M. C., Ripoll, A., Pérez, N., Prévôt, A. S. H., Canonaco, F., Querol, X., and Alastuey, A.: Chemical characterization of submicron regional background aerosols in the



western Mediterranean using an Aerosol Chemical Speciation Monitor, Atmos. Chem. Phys., 15, 6379–6391, doi: 10.5194/acp-15-6379-2015, 2015.

Minguillón, M. C., Pérez, N., Marchand, N., Bertrand, A., TemimeRoussel, B., Agrios, K., Szidat, S., van Drooge, B. L., Sylvestre, A., Alastuey, A., Reche, C., Ripoll, A., Marco, E., Grimalt, J. O., and Querol, X.: Secondary organic aerosol origin in an urban environment. Influence of biogenic and fuel combustion precursors, Faraday Discuss., 189, 337–359, DOI: 10.1039/c5fd00182j, 2016

Miyazaki, Y., Kondo, Y., Takegawa, N., Komazaki, Y., Fukuda, M., Kawamura, K., Mochida, M., Okuzawa, K., and Weber, R. J.: Time-resolved measurements of water-soluble organic carbon in Tokyo, J. Geophys. Res., 111, D23206, doi:10.1029/2006jd007125, 2006.

Miyoshi, A., Hatakeyama, S., and Washida, N.: OH radical-initiated photooxidation of isoprene: an estimate of global CO production, J. Geophys. Res.-Atmos., 99, 18779–18787, 1994.

Mohr, C., DeCarlo, P. F., Heringa, M. F., Chirico, R., Slowik, J. G., Richter, R., Reche, C., Alastuey, A., Querol, X., Seco, R., Peñuelas, J., Jiménez, J. L., Crippa, M., Zimmermann, R., Baltensperger, U., and Prévôt, A. S. H.: Identification and quantification of organic aerosol from cooking and other sources in Barcelona using aerosol mass spectrometer data, Atmos. Chem. Phys., 12, 1649–1665, doi:10.5194/acp-12-1649-2012, 2012

Myriokefalitakis, S., Tsigaridis, K., Mihalopoulos, N., Sciare, J., Nenes, A., Kawamura, K., Segers, A., and Kanakidou, M.: In-cloud oxalate formation in the global troposphere: a 3-D modeling study, Atmos. Chem. Phys., 11, 5761-5782, doi:10.5194/acp-11-5761-2011, 2011.

Ng, N. L., Canagaratna, M. R., Zhang, Q., Jimenez, J. L., Tian, J., Ulbrich, I. M., Kroll, J. H., Docherty, K. S., Chhabra, P. S., Bahreini, R., Murphy, S. M., Seinfeld, J. H., Hildebrandt, L., Donahue, N. M., DeCarlo, P. F., Lanz, V. A., Prevôt, A. S. H., Dinar, E., Rudich, Y., and Worsnop, D. R.: Organic aerosol components observed in Northern Hemispheric datasets from Aerosol Mass Spectrometry, Atmos. Chem. Phys., 10, 4625– 4641, doi:10.5194/acp-10-4625-2010, 2010a

Ng, N. L., Canagaratna, M. R., Jimenez, J. L., Zhang, Q., Ulbrich, I. M., and Worsnop, D. R., Real-time methods for estimating organic component mass concentrations from aerosol mass spectrometer data, Environ. Sci. Tech., 45(3), 910-916, 2010b

Ng, N. L., Herndon, S. C., Trimborn, A., Canagaratna, M. R., Croteau, P. L., Onasch, T. B., Sueper, D., Worsnop, D. R., Zhang, Q., Sun, Y. L., and Jayne, J. T.: An Aerosol Chemical Speciation Monitor (ACSM) for Routine Monitoring of the Composition and Mass Concentrations of Ambient Aerosol, Aerosol Sci. Technol., 45, 780–794, doi: 10.1080/02786826.2011.560211, 2011

Norris, G. A., Vedantham, R., Wade, K., Brown, S., Prouty, J., and Foley, C.: EPA Positive Matrix Factorization (PMF) 3.0: fundamentals & user guide, U.S. Environmental Protection Agency, 2008.

Orsini, D. A., Ma, Y., Sullivan, A., Sierau, B., Baumann, K., and Weber, R.: Refinements to the Particle-Into-Liquid Sampler (PILS) for ground and airborne measurements of water soluble aerosol composition, Atmos. Environ., 37, 1243–1259, 2003.





Paatero, P. and Tapper, U.: Positive MAtrix Factorization : anon-negative factor model with optimal utilization of error estimates of data values. Environmetrics, 5, 111-126, 1994

Paatero, P.:A weighted non-negative least squares algorithm for three-way 'PARAFAC' factor analysis. Chemometrics and Intelligent Laboratory Systems, 38, (2), 223-242, 1997

Paatero, P.: The multilinear engine – A table-driven, least squares program for solving multilinear problems, including the n-way parallel factor analysis model, J. Comp. Graph. Stat., 8, 854–888, doi: 10.2307/1390831, 1999.

Park, J.-H., Goldstein, A. H., Timkovsky, J., Fares, S., Weber, R., Karlik, J., and Holzinger, R.: Eddy covariance emission and deposition flux measurements using proton transfer reaction – time of flight – mass spectrometry (PTR-TOF-MS): comparison with PTR-MS measured vertical gradients and fluxes, Atmos. Chem. Phys., 13, 1439–1456, doi:10.5194/acp-13-1439-2013, 2013

Parrish, D. D., Hahn, C. J., Williams, E. J., Norton, R. B., Fehsenfeld, F. C., Singh, H. B., Shetter, J. D., Gandrud, B. W., and Ridley, B. A.: Indications of photochemical histories of Pacific air masses from measurements of atmospheric trace species at Point Arena, California, J. Geophys. Res., 97, 15883–15901, 1992.

Parrish, D. D., Stohl, A., Forster, C., Atlas, E. L., Blake, D. R., Goldan, P. D., Kuster, W. C., and de Gouw, J. A.: Effects of mixing on evolution of hydrocarbon ratios in the troposphere, J. Geophys. Res.-Atmos., 112, D10S34, doi:10.1029/2006JD007583, 2007.

Parworth, C., Fast, J., Mei, F., Shippert, T., Sivaraman, C., Tilp, A., Watson, T., and Zhang, Q.: Long-term measurements of submicrometer aerosol chemistry at the Southern Great Plains (SGP) using an Aerosol Chemical Speciation Monitor (ACSM), Atmos. Environ. 106, 43–55, 2015.

Peltier, R. E., Weber, R. J., and Sullivan, A. P.: Investigating a Liquid-Based Method for Online Organic Carbon Detection in Atmospheric Particles, Aerosol Sci. Technol., 41, 1117–1127, doi: 10.1080/02786820701777465, 2007.

Petit, J.-E., Favez, O., Sciare, J., Crenn, V., Sarda-Estève, R., Bonnaire, N., Mocnik, G., Dupont, J.-C., Haeffelin, M., and Leoz-Garziandia, E.: Two years of near real-time chemical composition of submicron aerosols in the region of Paris using an Aerosol Chemical Speciation Monitor (ACSM) and a multi-wavelength Aethalometer, Atmos. Chem. Phys., 15, 2985–3005, doi: 10.5194/acp-15-2985-2015, 2015

Pope, C. A. and Dockery, D. W.: Health effects of fine particulate air pollution: Lines that connect, J. Air Waste Manage. Assoc., 56, 709–742, 2006

Reissell, A., Harry, C., Aschmann, S. M., Atkinson, R., and Arey, J.: Formation of acetone from the OH radical- and O3-initiated reactions of a series of monoterpenes, J. Geophys. Res., 104, 13869– 13879, 1999

Ripoll, A., Minguillón, M. C., Pey, J., Jimenez, J. L., Day, D. A., Sosedova, Y., Canonaco, F., Prévôt, A. S. H., Querol, X., and Alastuey, A.: Long-term real-time chemical characterization of submicron aerosols at Montsec (southern Pyrenees, 1570 m a.s.l.), Atmos. Chem. Phys., 15, 2935–2951, doi: 10.5194/acp-15-2935-2015, 2015.



Roukos, J., Plaisance, H., Leonardis, T., Bates, M., and Locoge, N.: Development and validation of an automated monitoring system for oxygenated volatile organic compounds and nitrile compounds in ambient air, J. Chroma. A, 1216, 8642–8651, doi:10.1016/j.chroma.2009.10.018, 2009.

Rudolph, J. and Johnen, F. J.: Measurements of light atmospheric hydrocarbons over the Atlantic in regions of low biological activity, J. Geophys. Res., 95, 20583–20591, doi:10.1029/JD095iD12p20583, 1990

Sauvage, S., Plaisance, H., Locoge, N., Wroblewski, A., Coddeville, P., and Galloo, J. C.: Long term measurement and source apportionment of non-methane hydrocarbons in three French rural areas, Atmos. Environ., 43, 2430–2441, doi:10.1016/j.atmosenv.2009.02.001, 2009

Sciare, J., Cachier, H., Oikonomou, K., Ausset, P., Sarda-Estève, R., and Mihalopoulos, N.: Characterization of carbonaceous aerosols during the MINOS campaign in Crete, July–August 2001: a multi-analytical approach, Atmos. Chem. Phys., 3, 1743-1757, doi:10.5194/acp-3-1743-2003, 2003.

Sciare, J., Oikonomou, K., Favez, O., Liakakou, E., Markaki, Z., Cachier, H., and Mihalopoulos, N.: Long-term measurements of carbonaceous aerosols in the Eastern Mediterranean: evidence of long-range transport of biomass burning, Atmos. Chem. Phys., 8, 5551-5563, doi:10.5194/acp-8-5551-2008, 2008.

Sciare, J., d'Argouges, O., Sarda-Esteve, R., Gaimoz, C., Dolgorouky, C., Bonnaire, N., Favez, O., Bonsang, B., and Gros, V.: Large contribution of water-insoluble secondary organic aerosols in the region of Paris (France) during wintertime, J. Geophys. Res.-Atmos., 116, D22203, doi: 10.1029/2011JD015756, 2011.

Seibert, P., Kromp-Kolb, H., Baltensperger, U., Jost, D. T., Schwikowski, M., Kasper, A. and Puxbaum, H.: Trajectory analysis of aerosol measurements at high alpine sites. EUROTRAC Symposium 94. Garmish-Partenkirchen, Germany, Academic Publishing BV The Hague, 1994.

Seinfeld, J. H., Pandis, S. N.: Atmos. Chem. and Phys., edited by: Wiley-Interscience, New York, 1998.

Simpson, I. J., Akagi, S. K., Barletta, B., Blake, N. J., Choi, Y., Diskin, G. S., Fried, A., Fuelberg, H. E., Meinardi, S., Rowland, F. S., Vay, S. A., Weinheimer, A. J., Wennberg, P. O., Wiebring, P., Wisthaler, A., Yang, M., Yokelson, R. J., and Blake, D. R.: Boreal forest fire emissions in fresh Canadian smoke plumes: C1-C10 volatile organic compounds (VOCs), $CO_2$, CO, $NO_2$, NO, HCN and $CH_3CN$, Atmos. Chem. Phys., 11, 6445–6463, doi:10.5194/acp-11-6445-2011, 2011.

Slowik, J. G., Vlasenko, A., McGuire, M., Evans, G. J., and Abbatt, J. P. D.: Simultaneous factor analysis of organic particle and gas mass spectra: AMS and PTR-MS measurements at an urban site, Atmos. Chem. Phys., 10, 1969–1988, doi:10.5194/acp-10-1969- 2010, 2010.

Sorooshian, A., Brechtel, F. J., Ma, Y. L., Weber, R. J., Corless, A., Flagan, R. C., and Seinfeld, J. H.: Modeling and characterization of a particle-into-liquid sampler (PILS), Aerosol Sci. Tech., 40, 396–409, 2006



Sorooshian, A., Lu, M. L., Brechtel, F. J., Jonsson, H., Feingold, G., Flagan, R. C., and
Seinfeld, J. H., On the source of organic acid aerosol layers above clouds, Environ. Sci.
Technol., 41(13), 4647-4654, 2007
Stein, A., Draxler, R., Rolph, G., Stunder, B., Cohen, M. and Ngan, F.: NOAA's HYSPLIT
atmospheric transport and dispersion modeling system., Bull. Amer. Meteor. Soc.,
doi:10.1175/BAMS-D-14-00110.1, 2015
Stohl, A.: Trajectory statistics-a new method to establish sourcereceptor relationships of air
pollutants and its application to the transport of particulate sulfate in Europe, Atmos.
Environ., 30, 579–587, 1996.
Sullivan, A. P., Weber, R. J., Clements, A. L., Turner, J. R., Bae, M. S., and Schauer, J. J.: A
method for on-line measurement of water-soluble organic carbon in ambient aerosol particles:
Results from an urban site, Geophys. Res. Lett., 31, L13105, doi:10.1029/2004GL019681,
13  2004

Sullivan, A. P., Peltier, R. E., Brock, C. A., de Gouw, J. A., Holloway, J. S., Warneke, C.,
Wollny, A. G., and Weber, R. J.: Airborne measurements of carbonaceous aerosol soluble in
water over northeastern United States: Method development and an investigation into water-
soluble organic carbon sources, J. Geo-phys. Res.-Atmos., 111, D23S46, doi:
10.1029/2006JD007072, 2006.
Thunis, P., Triacchini, G., White, L., Maffeis, G., Volta, V.: Air pollution and emission
reductions over the Po-valley: Air Quality Modelling and Integrated Assessment, 18th world
IMACS Congress and MODSIM09 International Congress on Modeling and Simulation,
Interfacing Modeling and Simulation with Mathematical and Computational Sciences, pp.
2335–234, 2009
Tian, Y. Z., Shi, G. L., Han, S. Q., Zhang, Y. F., Feng, Y. C., Liu, G. R., Gao, L. J., Wu, J. H.,
and Zhu, T.: Vertical characteristics of levels and potential sources of water-soluble ions in
PM10 in a Chinese megacity, Sci. Total Environ., 447, 1–9, 2013
Tyndall, G .S., Cox, R. A., Granier, C., Lesclaux, R., Moortgat, G. K., Pilling, M. J.,
Ravishankara, A. R., and Wallington, T. J.: Atmospheric chemistry of small organic peroxy
radicals, J. Geophys. Res., 106, 12157–12182, 2001.
Ulbrich, I. M., Canagaratna, M. R., Zhang, Q., Worsnop, D. R., and Jimenez, J. L.:
Interpretation of organic components from Positive Matrix Factorization of aerosol mass
spectrometric data, Atmos. Chem. Phys., 9, 2891-2918, doi:10.5194/acp-9-2891-2009, 2009.
Vlasenko, A., Slowik, J. G., Bottenheim, J. W., Brickell, P. C., Chang, R. Y. W., Maedonald,
A. M., Shantz, N. C., Sjostedt, S. J., Wiebe, H. A., Leaitch, W. R., and Abbatt, J. P. D.:
Measurements of VOCs by proton transfer reaction mass spectrometry at a rural Ontario site:
Sources and correlation to aerosol composition, J. Geophys. Res., 114, D21305,
doi:10.1029/2009JD012025, 2009
Volkamer, R., San Martini, F., Molina, L. T., Salcedo, D., Jimenez, J. L., and Molina, M. J.,
A missing sink for gas-phase glyoxal in Mexico City: Formation of secondary organic
aerosol, Geophys. Res. Let., 34(19), 2007



Warneck, P., In-cloud chemistry opens pathway to the formation of oxalic acid in the marine atmosphere, Atmos. Environ., 37(17), 2423-2427,2003

Weber, R. J., Sullivan, A. P., Peltier, R. E., Russell, A., Yan, B., Zheng, M., de Gouw, J., Warneke, C., Brock, C., Holloway, J. S., Atlas, E. L., and Edgerton, E.: A study of secondary organic aerosol formation in the anthropogenic-influenced southeastern United States, J. Geophys. Res., 112, D13302, doi: 10.1029/2007jd008408, 2007.

Wisthaler, A., Jensen, N. R., Winterhalter, R., Lindinger, W., and Hjorth, J.: Measurements of acetone and other gas phase product yields from the oh-initiated oxidation of terpenes by protontransfer-reaction mass spectrometry (ptr-ms), Atmos. Environ., 35, 6181–6191, 2001.

Yuan, B., Shao, M., de Gouw, J., Parrish, D. D., Lu, S., Wang, M., Zeng, L., Zhang, Q., Song, Y., Zhang, J., and Hu, M.: Volatile organic compounds (VOCs) in urban air: how chemistry affects the interpretation of positive matrix factorization (PMF) analysis, J. Geophys. Res., 117, D24302, 1–17, doi:10.1029/2012JD018236, 2012.

Zannoni, N., Dusanter, S., Gros, V., Sarda Esteve, R., Michoud, V., Sinha, V., Locoge, N., and Bonsang, B.: Intercomparison of two comparative reactivity method instruments inf the Mediterranean basin during summer 2013, Atmos. Meas. Tech., 8, 3851-3865, doi:10.5194/amt-8-3851-2015, 2015.

Zannoni, N., Gros, V., Sarda Esteve, R., Kalogridis, C., Michoud, V., Dusanter, S., Sauvage, S., Locoge, N., Colomb, A., and Bonsang, B.: Summertime OH reactivity from a receptor coastal site in the Mediterranean basin, Atmos. Chem. Phys. Discuss., doi:10.5194/acp-2016-684, in review, 2016.

Zhang, Q., Jimenez, J., Canagaratna, M., Ulbrich, I., Ng, N., Worsnop, D., and Sun, Y.: Understanding atmospheric organic aerosols via factor analysis of aerosol mass spectrometry: a review, Anal. Bioanal. Chem., 401, 3045–3067, 2011.





1   Table 1: Summary of VOC measurements performed at Cape Corsica during the ChArMEx
2   SOP2 field campaign. DL stems for Detection Limit

| Instrument | Time resolution | # species | # species > DL | Overall uncertainties (%) | DL range (ppt) | Examples | Mean ± 1σ (ppt) | DL (ppt) |
|---|---|---|---|---|---|---|---|---|
| PTR-ToFMS | 10 min | 16 | 16 | 6-23 | 7-500 | isoprene | 194 ± 224 | 20 |
| | | | | | | sum terpenes | 407 ± 462 | 15 |
| | | | | | | acetaldehyde | 329 ± 118 | 50 |
| | | | | | | acetic acid | 1152 ± 405 | 110 |
| online GC/FID-FID | 90 min | 43 NMHC<br><br>C2-C12 | 22 | 5-23 | 10-100 | ethane | 891 ± 187 | 50 |
| | | | | | | butane | 65 ± 92 | 20 |
| | | | | | | propene | 31 ± 13 | 10 |
| | | | | | | ethyne | 92 ± 49 | 20 |
| | | | | | | benzene | 27 ± 12 | 10 |
| | | | | | | toluene | 77 ± 65 | 20 |
| Online GC/FID-MS | 90 min | 16 OVOCs<br><br>C3-C7<br>6 NMHCS | 22 | 5-14 | 5-100 | α-pinene | 108 ± 77 | 10 |
| | | | | | | B-Pinene | 141 ± 124 | 10 |
| | | | | | | limonene | 31 ± 35 | 10 |
| | | | | | | ethanol | 184 ± 79 | 20 |
| | | | | | | hexanal | 101 ± 50 | 20 |
| Offline solid adsorbants | 180 min | 35 NMHCs<br><br>C5-C16<br>5 C6-C12 n-aldehydes | 28 | ~25 | ~5 | nonane | 8 ± 46 | 5 |
| | | | | | | decane | 3 ± 3 | 5 |
| | | | | | | styrene | 6 ± 5 | 5 |
| | | | | | | hexanal | 17 ± 13 | 5 |
| Offline DNPH | 180 min | 16<br>C1-C8 | 14 | ~25 | 6-40 | formaldehyde | 2483 ± 868 | 40 |
| | | | | | | acetone | 3430 ± 1126 | 20 |
| | | | | | | MEK | 481 ± 385 | 20 |
| | | | | | | MACR | 59 ± 35 | 15 |
| | | | | | | GLY | 146 ± 81 | 15 |



Table 2 : Back-trajectory clusters for the ChArMEx SOP2 field campaign in Cape Corsica.
The averaged transport time corresponds to the time spent since the last anthropogenic
contamination, i.e. since the air masses left the continental coasts.

| Clusters | Source Region | Averaged transport Time | Contribution (%) |
|---|---|---|---|
| Marine West | South France, North East Spain | 36 h->48 h | 30% |
| Europe-North East | North Italy | 10 h-20 h | 26% |
| Corsica South | Corsica, Sardinia | 12 h-24 h* | 8% |
| France-North West | South East france | 12 h-18 h | 6% |
| Calm-Low wind | Local | Not applicable | 30% |

*For the Corsica-South cluster, the transport time corresponds to the time spent by the air
masses above land (Corsica and Sardinia Islands) before flying over the sea





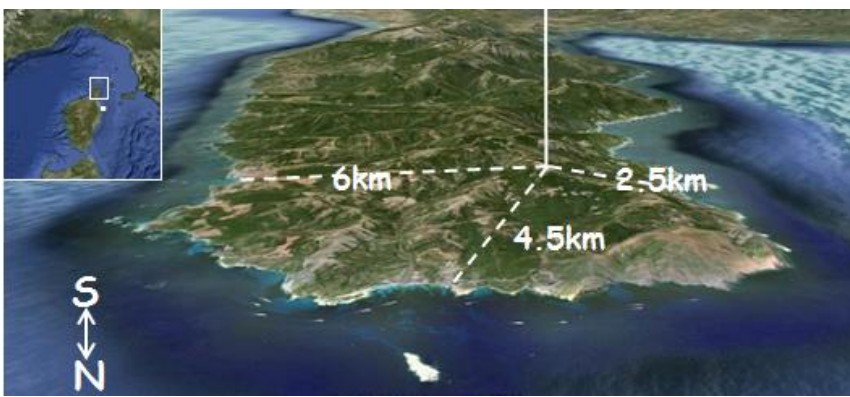

2 Figure 1: Localisation and geographical configuration of the measurement site at ERSA in
3 Cape Corsica (source: google map). The white solid square in the insert (top left) represents
4 the localisation of the city of Bastia.




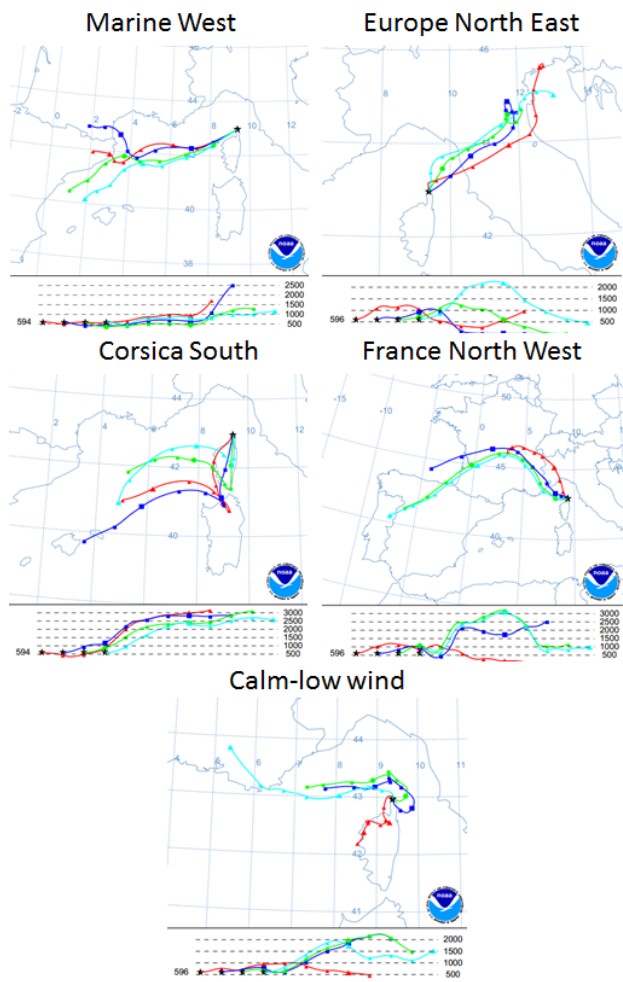

Figure 2: Five Back-trajectory clusters identified for the ChArMEx SOP2 field campaign at
Cape Corsica. This classification was conducted using back-trajectories calculated by the
HYSPLIT Model (NOAA-ARL). The five clusters are illustrated by example maps for 4
trajectories (interval of 6 hours between each, time of arrival indicates by different colours of
trajectory) for five single days representatives of an isolated cluster (07/25, 07/21, 07/28,
07/30 and 07/18 for Marine West, Europe-North East, Corsica South, France-West and Calm-
low wind, respectively).



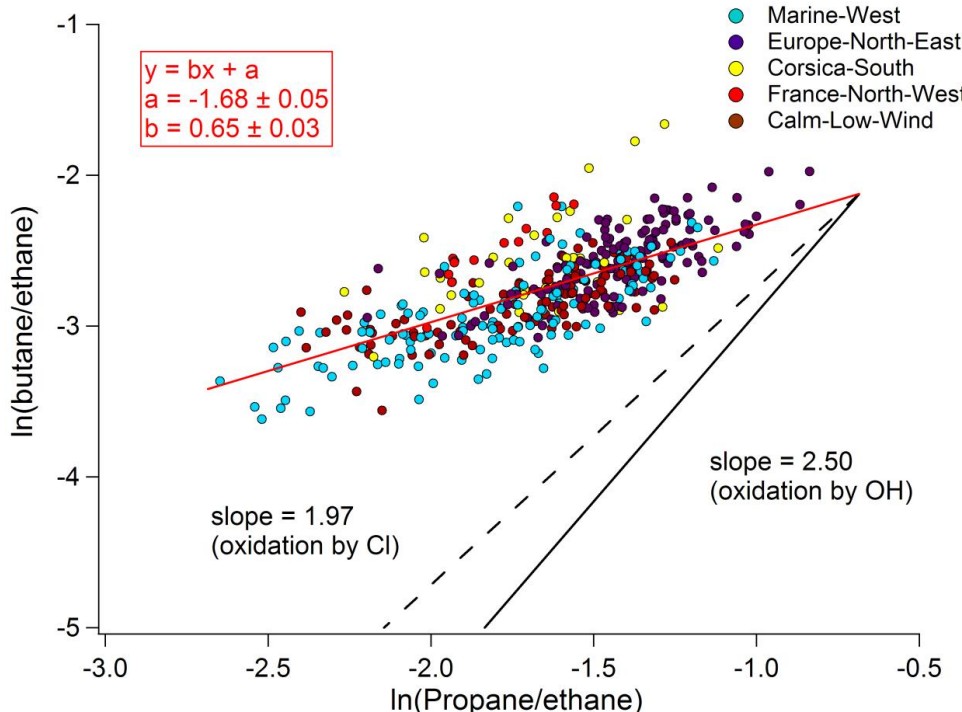

Figure 3 : Evolution of Ln(butane/ethane) as a function of Ln(propane/ethane) during the
ChArMEx SOP2 field campaign. The data were color-coded as a function of the back-
trajectory clusters (light blue, purple, yellow, red and brown for the Marine-West, Europe-
North-East, Corsica-South, France-North-West and Calm-Low-Wind clusters, respectively).
The red line corresponds to the linear regression. Black lines correspond to the theoretical
kinetic evolution of the ratios of alkanes due to oxidation by OH only (solid line) or Cl only
(dashed line).









Figure 4: Time series of selected trace gases and wind direction at cape Corsica during the
ChArMEx SOP2 field campaign. The coloured areas correspond to back-trajectory clusters
(light blue, purple, yellow, pink and orange-brown for the Marine-West, Europe-North-East,
Corsica-South, France-North-West and Calm-Low-Wind clusters, respectively).





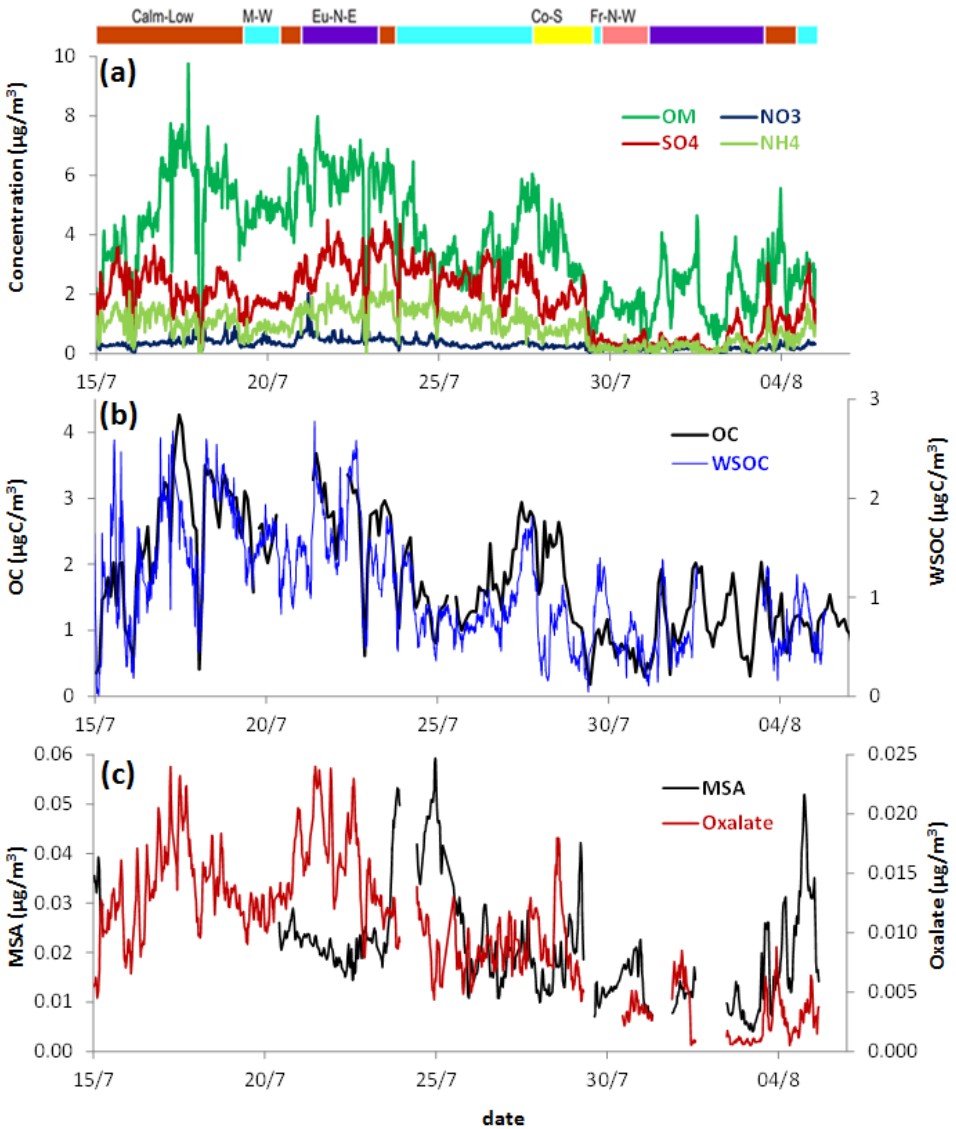

Figure 5: Temporal variability at Cape Corsica of (a) submicron (NR-PM$_1$) chemical constituents measured by ACSM, (b) OC (PM$_{2.5}$) and WSOC (PM$_1$) measured by OCEC Sunset Field instrument and PILS-TOC, (c) MSA and oxalate (PM$_{10}$) measured by PILS-IC. The coloured areas at the top correspond to back-trajectory clusters (light blue, purple, yellow, pink and orange-brown for the Marine-West (M-W), Europe-North-East (Eu-N-E), Corsica-South (Co-S), France-North-West (Fr-N-W) and Calm-Low-Wind (Calm-Low) clusters, respectively).







Figure 6: Time series for the contribution of the 6 gas-phase PMF factors together with
Temperature, CO, the measured Organic fraction of aerosols, and wind speed. The coloured
areas correspond to back-trajectory clusters (light blue, purple, yellow, pink and orange-
brown for the Marine-West (M-W), Europe-North-East (Eu-N-E), Corsica-South (Co-S),
France-North-West (Fr-N-W) and Calm-Low-Wind clusters, respectively).



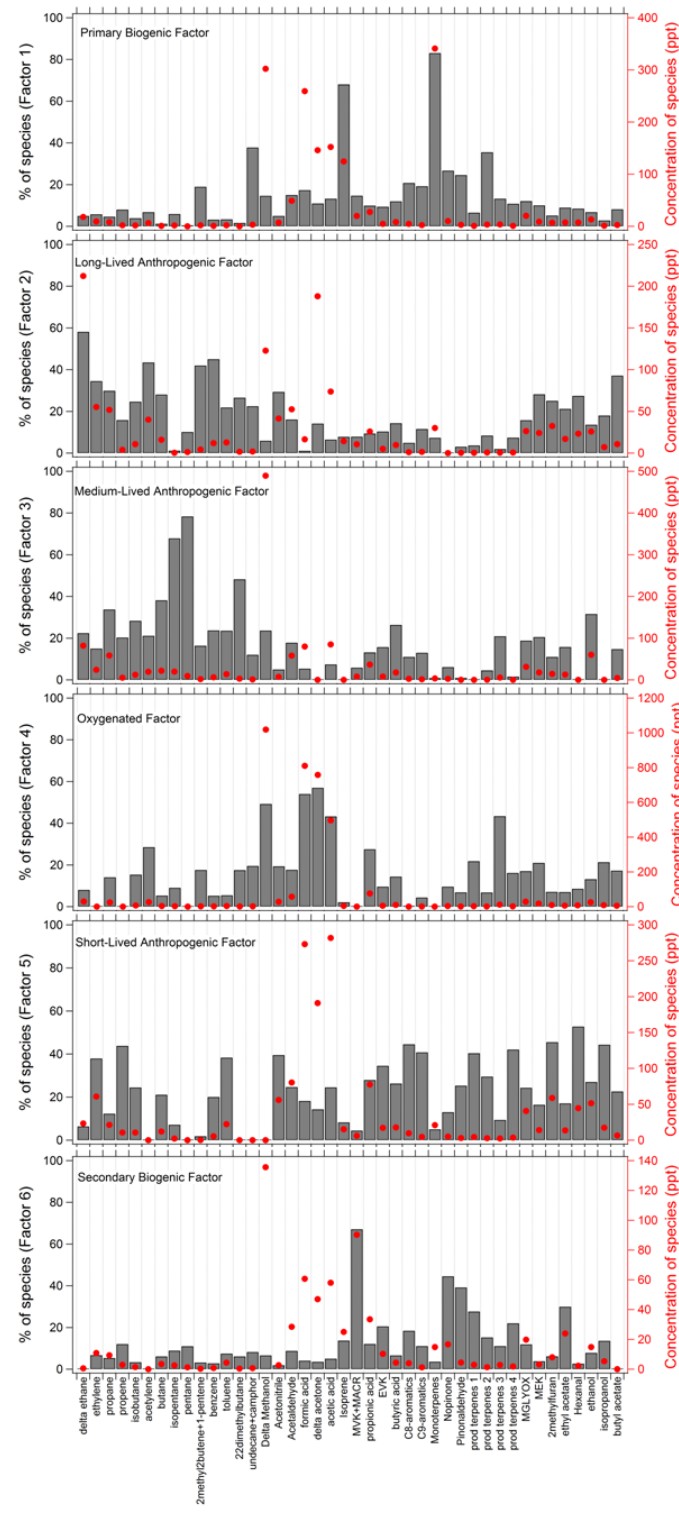





Figure 7: Profiles of the 6 gas-phase PMF factors, with contributions of the factors to each
species (black histograms, left axis in %) and contribution of the species to each factor (red
circles, right axis in ppt). The "prod terpenes" 1, 2, 3 and 4 corresponds to the m/z 99, 111,
113 and 155 signals from the PTR-ToFMS measurements, respectively, which have been
attributed to oxidation products of terpenes (Holzinger et al., 2005; Lee et al., 2006; Vlasenko
et al., 2009; Fares et al., 2012; Park et al., 2013).



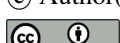

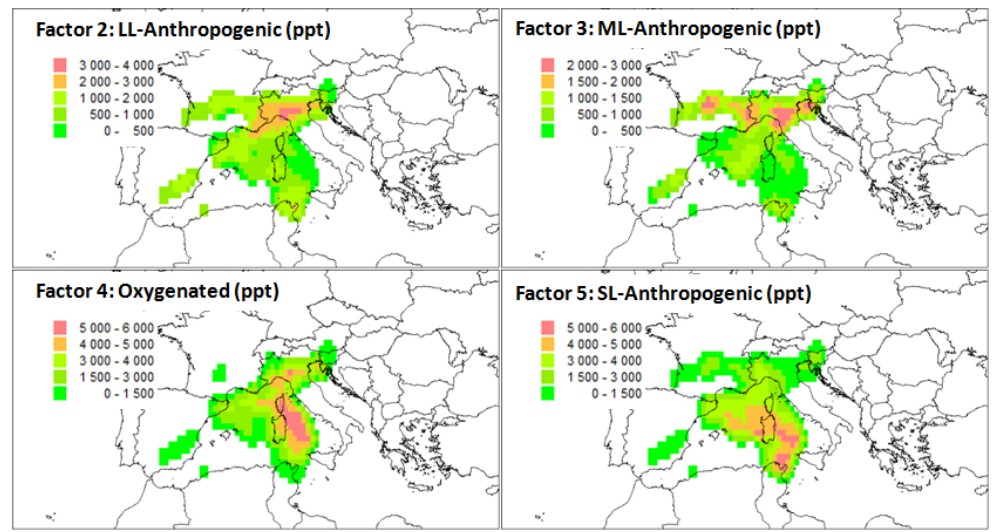

2 Figure 8: Source identification for the 6 gas-phase PMF factors, using the CF model.
3 Contributions are in units of ppt.





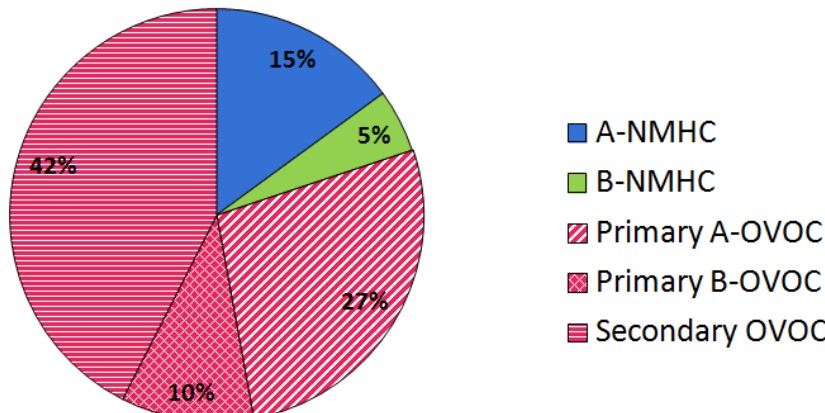

Figure 9: Distribution of the different VOC groups (ANMHC: Anthropogenic NMHCs (blue),
BNMHC: Biogenic NMHCs (green), OVOC: Oxygenated VOCs (pink)), calculated from the
database used for PMF analysis (same as bottom panel of Fig. S5). The OVOC group is
divided into three sub-classes to account for their different origins: Primary anthropogenic
(Primary A-OVOC, diagonal stripes), Primary biogenic (Primary B-OVOC, grid pattern) and
secondary origin from the oxidation of both anthropogenic and biogenic VOCs (Secondary
OVOC, horizontal stripes). The partitioning of these OVOCs into the three sub-classes is
described in section 4.2.4.



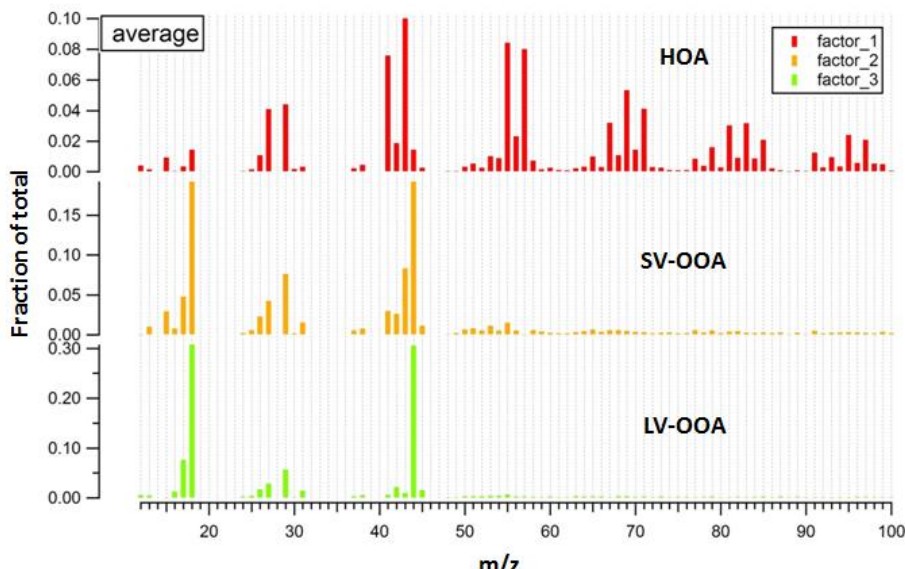

2    Figure 10 : Mass spectra profile obtained for the 3 factor constrained PMF solution (factor 1 =
3    HOA (red); factor 2 = SV-OOA (orange); factor 3 = LV-OOA (green)).





2     Figure 11 : Time-series of: (a) HOA (black) with Black Carbon (grey), (b) SV-OOA (black)
3     with WSOC (grey), (c) LV-OOA (black) with oxalate (grey).





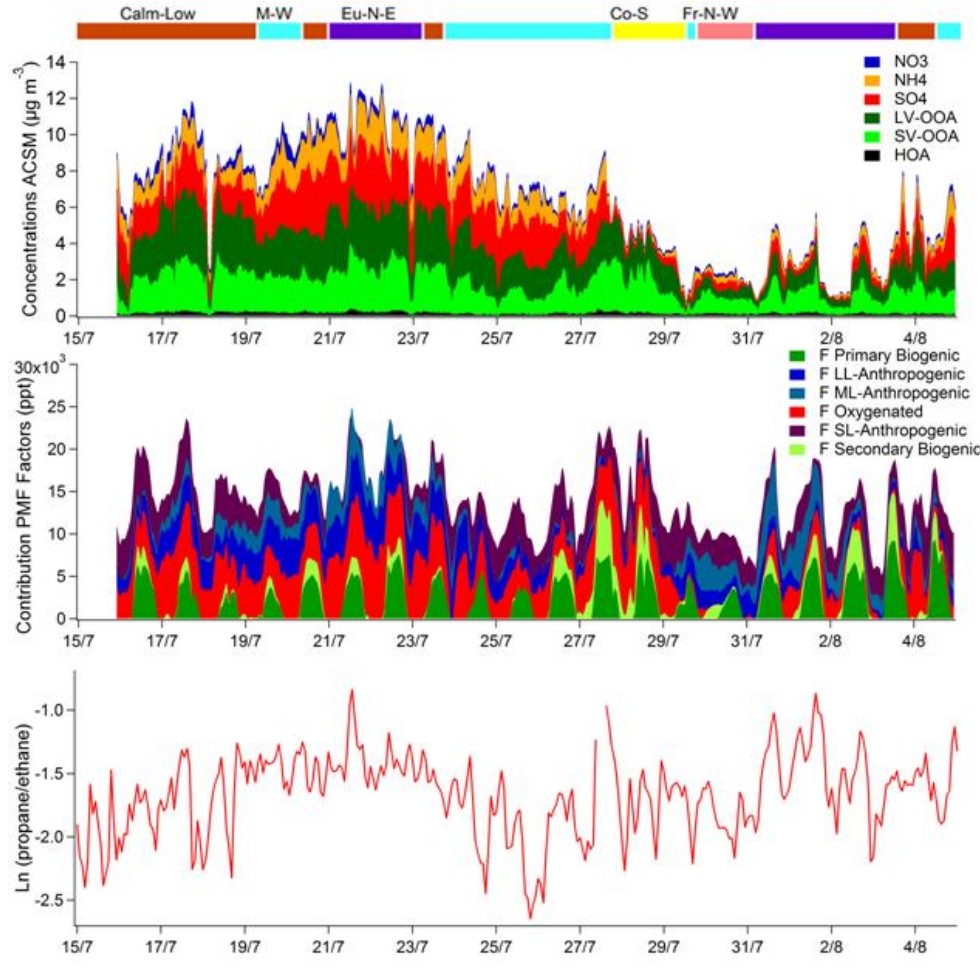

Figure 12 : Stacked time series of aerosol fractions (top panel), of VOC PMF Factors (middle
panel), and of Ln(propane/ethane) as a proxy for photochemical age (bottom panel). F-LL, F-
ML and F-SL-Anthropogenic refer to the Long-Lived, Medium-Lived and Short-Lived
Anthropogenic factors, respectively. Coloured areas at the top correspond to back-trajectory
clusters (light blue, purple, yellow, pink and orange-brown for the Marine-West (M-W),
Europe-North-East (Eu-N-E), Corsica-South (Co-S), France-North-West (Fr-N-W) and Calm-
Low-Wind (Calm-Low) clusters, respectively).

