# Peer review of "Organic carbon at a remote site of the western 1 Mediterranean Basin: sources and chemistry during the 2 ChArMEx SOP2 field experiment 3"

_Atmospheric Chemistry and Physics, 2016_

## Referee Comment (RC1) · Anonymous Referee #1 · 7 Feb 2017

The manuscript deals with measurements of various different VOCs together with inorganic gases and aerosols on Corsica. The measurement period is fairly short, three weeks. This data is used for PMF modelling in order to detect source areas and atmospheric chemistry of the compounds. The manuscript is well written, but the measurements are not adequately described and the input data is not presented at all. The manuscript is suitable for publication in ACP after minor corrections and additions.

1. Online analysis using Perkin-Elmer instrument. The calibration gas contains also monoterpenes. Were they also analyzed although Nafion dryers are known to isomerize monoterpenes?

2. Online analysis with Markes air server. BVOCs and OVOCs were analyzed using Markes air server unit and KI ozone scrubber was used. What is meant by BVOCs here? Monoterpenes or isoprene are not included in the standard. Were they analyzed? It would be very interesting to see the comparison with monoterpenes analyzed with the off line technique.

KI scrubber is generally used when analyzing aldehydes and ketones. Have they been tested for BVOCs? I would be interested to see recoveries of monoterpenes. Diluting the sample probably increases blank value quite a lot. Was this taken into account when subtracting blank values? How much diluting increased the blanks?

3. Collection into adsorbent tubes. I think a short description of the analysis should be given here too. For the collection of 35 compounds (C5-C16) on the adsorbent tubes, air was passed through MnO2 ozone scrubber. How many MnO2 coated nets were used? What was the sampling efficiency? According to Calogirou et al (1999) some oxygen containing compounds (for example linalool) are lost in these scrubbers and Pollmann et al (2005) found sesquiterpenes total loss in MnO2 scrubbers.

Results and discussion

No measurement data is shown in the manuscript. Some statistics of the model input data should be given (mean, standard deviation). For many of the compounds at least two overlapping methods were used and they were also compared although the data is not shown. However, this comparison would be very useful for others conducting similar analysis at least if severe flaws are observed. Where all the data used for PMF modelling? When studying photochemical ages of air masses the authors observe much lower slope than theoretical (Fig.3) and they attribute this to mixing between air parcels of different histories and origins. There is probably quite heavy marine traffic around Corsica that can affect also VOC ratios. Would it be possible to remove ship emissions from marine sector according to high NOx or SO2 concentrations for

example? For PMF modelling inorganic gases were not used, neither inorganic ions, although they could have been analyzed with PILS. These analysis would have provided important addition to the PMF modelling. NOx would have given information of traffic and combustion, SO2 from ship emissions and for example K-ions of wood burning. Why did you leave inorganic compounds outside modelling? Did you try PMF modelling with inorganic compounds? Figure 8 in supplementary material is nice. I think it could be included in the manuscript. The diurnal cycles of biogenic factors with maxima during day indicate light dependent emissions. Also different diurnal cycles have been observed with higher concentrations at night due to efficient mixing and sink reactions during daytime. (Harrison et al., 2001; Hakola et al., 2012)

References: Calogirou A, 1996. Decomposition of terpenes by ozone during sampling on Tenax. Analytical Chemistry 68, 1499-1506. Hakola et al., 2012. In situ measurements of volatile organic compounds in a boreal forest. Atmos. Chem. Phys., 12, 11665-11678. Harrison et al., 2001. Ambient isoprene and monoterpene concentrations in a Greek fir (Abies Borisii-regis) forest. Reconciliation with emissions measurements and effects on measured OH concentrations. Atmospheric Environment, 4699-4711. Pollmann et al., 2005. Analysis of atmospheric sesquiterpenes: Sampling losses and mitigation of ozone interferences, Environmental Science & Technology, 9620-9629, DOI: 10.1021/es050440w

---

## Referee Comment (RC2) · Anonymous Referee #2 · 19 Feb 2017

This paper deals with the interpretation of a comprehensive set of data on the organics (aerosol and gaseous phases) recorded during an intensive sampling period in the Cap Corsica remote background site. The article is well written. Results presented are of interest for the scientific community. Sources of organic carbon aerosols are investigated by simultaneously applying factor analysis to the gas and aerosol species determined during a very short period (20 days in summer 2013) at a monitoring site in Corsica. A similar methodology was applied in previous works such as those by Crippa et al 2013a, and / or Slowik 2010. In the present paper the source apportionment results are combined with meteorological analysis for identifying possible source

areas. As mentioned in the text, 14C was also analyzed during the field campaign. Definitely, the results on 14C can provide very valuable information for the objective of the present paper. These analyzes are complementary to the techniques used in the present study and will allow a better understanding of the sources and chemistry of the organics. Thus, authors are referring to the results obtained by the 14C analysis in the results section (page 32, 24-29) and highlight them in the conclusions section (page 35; 8-11). Would the authors consider including the 14C data in the present paper? This can greatly increase the quality of the paper. However, if the authors consider that the results of 14C should be presented and discussed in another article, then the sentences relating to this unpublished material should be removed from the conclusions section.

The title should be modified. The use of composition and chemistry is redundant. "

Minor corrections

Abstract: I recommend authors to rewrite and reorder the abstract. I suggest deleting "First" (L7) and "second" L15). Line 9: Therefore? L12-13: last sentence of the first paragraph . Please, indicate the average concentration of the non-refractory submicron fraction. Do you have data on PM1? L27: ACSM should be mentioned earlier (L12-13) when contribution of OM to the non-refractory mass is described L28: Do you mean that 96% of the OVOCs are associated to SOA? INTRODUCTION

Page 4 L21: Use the acronym (HOA), already defined, instead of Hydrocarbon Organic Aerosol L21-L22: Please, indicate the acronyms for cooking aerosols and Biomass burning aerosols Page 5: L30 Why do you distinguish among "composition" and chemistry?

Methodology is described in two different sections (sections 2 and 3). In total is a very long part with a quite detailed description of the number of techniques and methods applied. The first methodological section, Section 2, is entitled "The ChArMEx experiment" . I think this title is not appropriated given that description is circumscribed to

the methods used in the present paper. The ChArMEx experiment is a large scale experiment covering different research fields in the Mediterranean. The present study is focused in the SOP2experiment, and I guess this SOP2 covers not only the Corsica measurements but also other measurements in different Mediterranean areas". Section 2 describes the instrumental technique used but also the back trajectory analysis and the estimation of the photochemical age of the air masses. Section 3 describes the source receptor models applied used (PMF, ME-2, CF) . Does it mean that back trajectory analysis form part of the ChArMEx experiment but the PMF does not? I think these two sections should be unified in a Methodology section or organised following a different structure: measurements (gaseous –online, offline., aerosols),data treatment, models. OC EC online: which thermos optical protocol was used? Were measurements compared with off line determinations? Were measurements of PM1 available? Could you compare the NR PM1 levels obtained by ACSM with optical or gravimetrical measurements of PM1? Measurement of BC by means of a AE-31 were available. However, these measurements were not used in the paper. Data on BC, could be added to the ACSM data and compared to the PM levels (if available. BC data should be comrade with the EC measurements.

RESULTS

In general, average concentrations of measured compounds/species should be compared with results obtained in other comparable areas P21, L32. Please, could you indicate what is the percentage of the NR-PM1 with respect to the total PM1? P26 L4: Solar radiation may also influence biogenic VOCs P26 L23-26. As shown in Figure 6 there is not a clear anti-correlation between Factor 6 and wind speed. Even, during specific days and increase in wind speed is related with an increase in Factor 6. Please, rewrite P30 L19-L20: Can the diurnal cycles with maxima at midday related to specific transport scenarios (breeze)?

P31 L25-34. Authors distinguish two periods with impact of processed anthropogenic / continental air masses: 19 - 27 of July and 30/07-03/08. However, as shown in Figure

12 there are marked differences between this two periods. However, the time evolution of both aerosol component s and VOC factors are very similar for the period 17 - 27 of July, including the calm period. The time evolution of ACSM compounds during this period 17 - 27 of July, could be related to the summer recirculation of air masses as shown in previous papers on ACSM data in the area (Minguillon et al Atmos. Chem. Phys., 15, 6379–6391, 2015, Ripoll et al, Atmos. Chem. Phys., 15, 2935–2951, 2015) and described in Pey et al, Atmos. Res., 94, 422–435, 2009, among others. These scenarios characterized by the recirculation of air masses could explain the mix of long, medium and short lived VOCs

Table1: Increase width of first column

---

## Author Comment (AC1) · 16 May 2017

The comment to both reviews was uploaded in the form of a supplement.

Please also note the supplement to this comment:
http://www.atmos-chem-phys-discuss.net/acp-2016-955/acp-2016-955-AC1-supplement.pdf

---

## Author Response (AR1)

First, we would like to thank the reviewers for their valuable comments on the manuscript. We did our best to address all the comments and summarized the changes made to the revised manuscript below. Furthermore, these changes have been highlighted in yellow in the revised manuscript.

**Response to referee #1:**

*1. Online analysis using Perkin-Elmer instrument. The calibration gas contains also monoterpenes. Were they also analyzed although Nafion dryers are known to isomerize monoterpenes?*

Alpha and Beta pinene were not analyzed by the Perkin-Elmer Instrument but were analyzed using the MArkes/Agilent TD/GC instrument. However, in this study, we only used the mixing ratio of the sum of monoterpenes measured by PTR-ToF-MS, for the data and the PMF analysis.

*2. Online analysis with Markes air server. BVOCs and OVOCs were analyzed using Markes air server unit and KI ozone scrubber was used. What is meant by BVOCs here? Monoterpenes or isoprene are not included in the standard. Were they analyzed? It would be very interesting to see the comparison with monoterpenes analyzed with the off line technique.*

Isoprene, alpha and beta pinenes as well as limonene were measured using this online GC instrument. While these species were not included in the Praxair cylinder used for frequent calibrations, they were included in a NPL cylinder, which was also used to perform calibrations on the Markes instrument at the beginning and the end of the campaign only.
For indication, comparisons of isoprene, alpha-pinene, beta-pinene and limonene measured by the Markes instrument and PTR-MS or off-line techniques are shown in Figure 1. Isoprene, and alpha-pinene show good agreement between these instruments, whereas beta-pinene and Limonene concentrations measured by off-line techniques display some discrepancies. This disagreement has been explained by a potential thermo-degradation observed for beta-pinene which can also have impacts on Limonene measurements

Nevertheless, only isoprene and sum of monoterpenes concentrations measured by PTR-ToF-MS have been considered in this study for data and PMF analysis. Therefore these comparisons have not been added to the revised manuscript.

[Figure]

Figure 1: Comparisons of Isoprene, alpha-pinene, beta-pinene and limonene mixing ratios measured by the markes instrument and PTR-ToF-MS or off-line techniques.

*KI scrubber is generally used when analyzing aldehydes and ketones. Have they been tested for BVOCs? I would be interested to see recoveries of monoterpenes. Diluting the sample probably increases blank value quite a lot. Was this taken into account when subtracting blank values? How much diluting increased the blanks?*

Zero Air was sampled to have blanks using the sampling device including the dilution. The blank was not increased. But the dilution decreases the sensitivity of the method and consequently the detection limit. The DL reported take into account this configuration.

Here KI scrubber has been used since the markes instrument was mainly dedicated to aldehydes and ketones measurements. Nevertheless, KI scrubber was tested for BVOCs and the recoveries (with and without scrubber) for most of the monoterpenes (with and without ozone) are quite good. A paper will be submitted soon with these results.

[Figure]

*Deviation relative to the same BVOC mixture measured without scrubber*

*3. Collection into adsorbent tubes. I think a short description of the analysis should be given here too. For the collection of 35 compounds (C5-C16) on the adsorbent tubes, air was passed through MnO2 ozone scrubber. How many MnO2 coated nets were used? What was the sampling efficiency? According to Calogirou et al (1999) some oxygen containing compounds (for example linalool) are lost in these scrubbers and Pollmann et al (2005) found sesquiterpenes total loss in MnO2 scrubbers.*

This paper is already rather long and we did not want to give too detailed description of each analytical method which has been described elsewhere already. This is why we refer to the paper from Detournay et al, 2011 to a more detailed description of this method in the manuscript. In this paper, the reader will find a description and performance tests of both the sampling system and the analytical part.
The scrubber used is a commercialized one produced by THERMO. It is composed of 12 copper nests impregnated with $MnO_2$ and assembled in a cylinder, equipped with 2 inlet and outlet particles filter.

*A. DETOURNAY, S. SAUVAGE, N. LOCOGE, V. GAUDION, T. LEONARDIS, I. FRONVAL, P. KALUZNY, J.-C. GALLOO, Development of a sampling method for the simultaneous monitoring of long chain alkanes, long*

*chain carbonyl compounds and monoterpenes in remote areas, Journal of Environmental Monitoring, Vol 13, N°4, pp 983-990, 2011. DOI: 10.1039/C0EM00354A.*

*Results and discussion*
*No measurement data is shown in the manuscript. Some statistics of the model input data should be given (mean, standard deviation). For many of the compounds at least two overlapping methods were used and they were also compared although the data is not shown. However, this comparison would be very useful for others conducting similar analysis at least if severe flaws are observed.*

Some measurement data is shown in the manuscript. Figure 4 presents time series of 8 different gaseous species and figure 5 presents the temporal variability of submicron chemical constituents of aerosols. Concerning the gaseous species, we made the choice to present only 8 different species in the figure for clarity. Furthermore, these 8 different species are characterized by different lifetimes and origins (Acetylene and CO as anthropogenic long lived compounds, sum of monoterpenes and isoprene as biogenic primary compounds, acetone as an oxygenated compound originating from both anthropogenic and biogenic sources, NO and $NO_2$ as short lived compounds emitted by traffic and ozone as a long-lived photochemically formed compound). Furthermore, some statistics are given in table 1 for 24 different species measured by the different analytical methods (e.g. isoprene, monoterpenes, acetaldehyde, acetic acid, ethane, butane, propene, ethyne, benzene, toluene, alpha and beta-pinene etc…).
Following the comment of the referee, a table has been added in the revised supplementary material (table S5) with statistics (mean and standard deviation) of species used as input in the PMF analysis.
Concerning the comparisons of compounds measured by several methods we made the choice not to present them. Furthermore, the presentation of all the comparisons performed for this campaign would have led to tremendous quantity of figures and would have moved the paper away from the main focus. Therefore, these comparisons were only mentioned in the manuscript to inform the readers about the internal quality insurance procedure made to validate and select the most relevant technique for each species (regarding reliability and resolution time).

*Were all the data used for PMF modelling?*

As mentioned in the original manuscript, the full database was not used for PMF analysis. This analysis was conducted on a dataset of 42 species only (P16, line 5). Some species were not included because of being below detection limit most of the time or exhibiting a too low signal/noise ratio (P16, line 11-12). Furthermore, offline measurements whose sampling was averaged over 3h were not included because of too low time resolution (P16, line7-9). As mentioned previously, the table presenting input data has been added in the supplementary material.

*When studying photochemical ages of air masses the authors observe much lower slope than theoretical (Fig.3) and they attribute this to mixing between air parcels of different histories and origins. There is probably quite heavy marine traffic around Corsica that can affect also VOC ratios. Would it be possible to remove ship emissions from marine sector according to high NOx or SO2 concentrations for example?*

Indeed, the slope obtained in Figure 3 is much lower than the theoretical one. This disagreement has often been observed in the literature (e.g. Parrish et al., 1992; McKeen et al.,

1996) but not in this proportion. This observed lack of concordance has been attributed by Parrish et al. (2007) to the mixing between air parcels of different histories and origins during long-range transport. However, we agree with the first referee that changes of the theoretical slope can occur if the air masses were enriched in new emissions from different sources, such as ship or marine emissions, during the transport. This point has been added in the revised manuscript as follow (P15, L23-25):

"A deviation from theoretical slope could also occur if the sampled air masses were enriched in new emissions from different sources, such as ship or marine emissions, during the transport."

Nevertheless, removing ship emissions from the marine sector would be very difficult since it can occur from all wind sectors the measurement site being surrounded by the sea. Furthermore, using $NO_x$ and/or $SO_2$ high concentrations to exclude data points impacted by ship emissions would be difficult since quite low $NO_x$ and $SO_2$ levels were measured all along the campaign (see Figure 1 of the manuscript and Figure 2 of this document).

[Figure]

Figure 2: Time series of $SO_2$ measured at Cape Corsica during the ChArMEx SOP2 field campaign

While the disagreement in slope value between observation and theory remains not fully explain and uncertain, differences in the absolute ratio between various wind sectors allow assessing if some air masses are more aged than others.

*For PMF modelling inorganic gases were not used, neither inorganic ions, although they could have been analyzed with PILS. These analyses would have provided important addition to the PMF modelling. NOx would have given information of traffic and combustion, SO2 from ship emissions and for example K-ions of wood burning. Why did you leave inorganic compounds outside modelling? Did you try PMF modelling with inorganic compounds?*

In this study, the objectives were to study the main drivers of organic carbon evolution in both the gaseous and aerosol phases. Therefore, PMF was only conducted using organic gaseous species as inputs in a first step. In this kind of exercises, inorganic trace gases are not included in the input database but can be used as external tracers to help identifying the factors. The $NO_x$ and $SO_2$ concentrations were used this way leading to unconvincing results (no significant correlation observed with any of the factors). Therefore, other external tracers were used to help identifying the factors (CO, BC, organic aerosol mass concentrations, etc…).

Then, a joint PMF analysis with organic gaseous species and both organic and inorganic aerosol measurements was attempted. However, it did not allow to satisfactorily apportion

aerosol measurements and led to weaker solutions than the ME-2 analysis for the organic fraction and to an isolated inorganic aerosol factor giving no additional information. So, we finally applied separate factorization analyses for organic gaseous species and aerosols without including inorganic trace gases or aerosol constituents.

*Figure 8 in supplementary material is nice. I think it could be included in the manuscript. The diurnal cycles of biogenic factors with maxima during day indicate light dependent emissions. Also different diurnal cycles have been observed with higher concentrations at night due to efficient mixing and sink reactions during daytime. (Harrison et al., 2001; Hakola et al., 2012)*

We made the choice to show diurnal variations of PMF factors in the supplementary material only to keep the paper as concise as possible. Letting this figure in supplementary material will not suppress the possibility for the reader to have a look to the diurnal profiles and we therefore prefer to let it there.

**Response to referee #2:**

*As mentioned in the text, 14C was also analyzed during the field campaign. Definitely, the results on 14C can provide very valuable information for the objective of the present paper. These analyzes are complementary to the techniques used in the present study and will allow a better understanding of the sources and chemistry of the organics. Thus, authors are referring to the results obtained by the 14C analysis in the results section (page 32, 24-29) and highlight them in the conclusions section (page 35; 8-11). Would the authors consider including the 14C data in the present paper? This can greatly increase the quality of the paper. However, if the authors consider that the results of 14C should be presented and discussed in another article, then the sentences relating to this unpublished material should be removed from the conclusions section.*

We think that the $^{14}$C analysis is indeed helpful to apportion the origin of organic carbon measured during this campaign which is one of the objectives of this study. Removing the reference to these results in our study would certainly weaken our analysis.
However, figure presenting $^{14}$C data cannot be added to the revised manuscript since a paper on multisite (Cape Corsica and Mallorca) chemical composition of aerosol with detailed $^{14}$C data analysis will be submitted soon, and could not be so if these results are published before. Therefore, we decide not to include the $^{14}$C data in this study but to let the reference to this analysis in the section 4.4. Nevertheless, we decide to remove the sentences relating to this $^{14}$C analysis in the conclusion as suggested by the referee and we modify the last paragraph of the conclusion (P35, L11-17) as follow:

"A coupled analysis of VOC and OA sources was conducted. During biogenic periods, 3 the aerosol composition was dominated by (secondary) OA indicating a substantial impact of BVOCs on aerosols composition, while during periods of long range transport of anthropogenic/continental emissions, the inorganic and organic fractions of 5 submicron aerosol were similar. During the whole campaign, low levels of Hydrogen-like OA 6 (HOA) were observed (<0.3 μg m-3), indicating a weak influence of primary anthropogenic 7 sources on OA"

*The title should be modified. The use of composition and chemistry is redundant. "*

The title has been modified in the revised manuscript as follow: "Organic carbon at a remote site of the western Mediterranean Basin: sources and chemistry during the ChArMEx SOP2 field experiment"

*Minor corrections:*
*Abstract: I recommend authors to rewrite and reorder the abstract. I suggest deleting "First" (L7) and "second" L15). Line 9: Therefore? L12-13: last sentence of the first paragraph.*

As proposed by the second referee, we deleted the following group of words in the abstract of the revised manuscript: "First" (P2, L7), "Second" (P2, L15), "Therefore" (P2, L9) and the last words of the last sentence of the first paragraph: "(55% of the total mass of non-refractory submicron aerosol on average)" (P2, L13-14).
We also removed the following sentence in the revised abstract:
"We combined a back-trajectory analysis and an estimation of photochemical age to characterize air mass origins and chemical processing times, which confirmed the remote nature of the site." (P2, L6-8).

*Please, indicate the average concentration of the non-refractory submicron fraction. Do you have data on PM1? L27: ACSM should be mentioned earlier (L12-13) when contribution of OM to the non-refractory mass is described*

Average concentration and standard deviation of the non-refractory submicron fraction have been added in the revised abstract (P2, L11-14):
"Organic Matter (OM) dominated the aerosol chemical composition, representing 55% of the total mass of Non-Refractory-PM1 on average (average of $3.74\pm1.80$ µg m$^{-3}$), followed by sulphate (27%, $1.83\pm1.06$ µg m$^{-3}$), ammonium (13%, $0.90\pm0.55$ µg m$^{-3}$), and nitrate (5%, $0.31\pm0.18$ µg m$^{-3}$)."
ACSM has been mentioned earlier in the first paragraph (P2, L5-7) of the revised abstract:
"At the same time an exhaustive description of the chemical composition of fine aerosols was performed, especially by Aerosol Chemical Speciation Monitor (ACSM) measurements."

*L28: Do you mean that 96% of the OVOCs are associated to SOA?*

We meant that 96% of OAs are SOAs. We modified the sentence as follow (P2, L27-28):
"highlighting the close link between OVOCs and organic aerosols, the latter being mainly associated (96%) to the secondary OA fraction"

*INTRODUCTION*

*Page 4 L21: Use the acronym (HOA), already defined, instead of Hydrocarbon Organic Aerosol L21-L22: Please, indicate the acronyms for cooking aerosols and Biomass burning aerosols Page 5: L30 Why do you distinguish among "composition" and chemistry?*

P4 L20: "Hydrocarbon Organic Aerosol" has been replaced by "HOA"
P4 L21: Acronyms have been added for cooking organic aerosol (COA) and biomass burning organic aerosol (BBOA).
P5 L30: The term "composition" has been removed in the revised manuscript.

*Methodology is described in two different sections (sections 2 and 3). In total is a very long part with a quite detailed description of the number of techniques and methods applied. The first methodological section, Section 2, is entitled "The ChArMEx experiment". I think this title is not appropriated given that description is circumscribed to the methods used in the present paper. The ChArMEx experiment is a large scale experiment covering different research fields in the Mediterranean. The present study is focused in the SOP2experiment, and I guess this SOP2 covers not only the Corsica measurements but also other measurements in different Mediterranean areas". Section 2 describes the instrumental technique used but also the back trajectory analysis and the estimation of the photochemical age of the air masses. Section 3 describes the source receptor models applied used (PMF, ME-2, CF). Does it mean that back trajectory analysis form part of the ChArMEx experiment but the PMF does not? I think these two sections should be unified in a Methodology section or organised following a different structure: measurements (gaseous –online, offline., aerosols),data treatment, models.*

The section 2 has been renamed: "The ChArMEx SOP2 ground base field experiment". This appellation only concerns the measurement made in Cape Corsica between mid-july and beginning of august 2013. In this section, we described the measurement techniques as well as the back-trajectory analysis and the estimation of photochemical age which allow a description of measurements performed but also general conditions encountered during the campaign. The specific section dedicated to source-receptor models (PMF, ME-2 and CF) concerns the general descriptions of these models and a description of the procedures applied in this study. There are only a few considerations of the campaign itself. This is why we decided to treat them separately.

*OC EC online: which thermos optical protocol was used? Were measurements compared with off line determinations?*

The $PM_{2.5}$ thermo-optical protocol used for the OCEC Sunset Field Instrument is the one by default which has been designed by the manufacturer to minimize the duration of the thermo-optical analysis (down to 15min) and instrument sensitivity.

On-line EC and OC measurements in $PM_{2.5}$ by the Sunset Field Instrument are intercompared in the figures below with off-line undenuded filter sampling performed by Low-vol (2.3 $m^3$/h) leckel sampler in $PM_1$ every 12h (00:00-12:00 UTC; 12:00-00:00 UTC) using Palflex Gelman 47 quartz fibre filters and a the EUSAAR II thermo-optical method.

[Figure]

[Figure]

[Figure]

[Figure]

Satisfactory comparison results were obtained for both OC and EC within the uncertainties of the on-line OCEC sunset field instrument (see manuscript) and the off-line measurements (e.g. no VOC denuder and uncertainties associated with the filter sampling and off-line thermo-optical analysis)

These results have been mentioned in the revised manuscript (P12, L20-22) as follows:
"These online EC and OC measurements were also intercompared with analysis from off-line filter sampling to check their reliability, leading to satisfactory agreement between both methods (see Supplementary Material Fig. S3a)."
The figure above has also been added to the supplementary material (Fig. S3a)

*Were measurements of PM1 available? Could you compare the NR PM1 levels obtained by ACSM with optical or gravimetrical measurements of PM1?*

No other gravimetric measurement of PM1 was available. Therefore, no comparison could be made.
On the other hand, as mentioned in the manuscript NR PM1 from ACSM was compared against PM1 data derived from SMPS (with a constant density of 1.4). This comparison is displayed below. Comparison was performed on the 25-min time resolution of the ACSM and for the period 15/07-04/08 (N = 3,055 data points).

[Figure]

[Figure]

A very satisfactory agreement was found between the two datasets with a slope close to 1.00. The results of this intercomparison have been reported in the supplementary information (Fig S3c)

*Measurement of BC by means of a AE-31 were available. However, these measurements were not used in the paper. Data on BC, could be added to the ACSM data and compared to the PM levels (if available. BC data should be compared with the EC measurements).*

BC data were used in the paper and are presented in original Fig. 11 (top panel) in the same time than HOA. Therefore, the consistency of the HOA factor has been checked by the comparison of its temporal variability together with BC time series. As said in the original manuscript (P32, L17) low levels of BC were encountered for the whole campaign (BC<0.9 µg m$^{-3}$).
Intercomparison between EC (online OCEC Sunset Field Instrument) and BC (Aethalometer AE31) are presented in the figures below for the period 15/07-04/08 based on the 2-h time resolution of the Sunset Field Instrument (total of 251 data points).

[Figure]

A very satisfactory agreement was found between the two datasets (r² = 0.70) with a slope close of 1.00.
The results of this intercomparison have been mentioned in the revised manuscript (P12, L22-24), as follows:
"EC online measurements were also compared to BC measurements from an Aethalometer, leading to satisfactory agreement (see Supplementary Material Fig. S3b)."
The figure above has also been added to the supplementary material (Fig S3b).

*RESULTS:*
*In general, average concentrations of measured compounds/species should be compared with results obtained in other comparable areas P21, L32.*

Comparisons with other studies have been added in section 4.1.2 of the revised manuscript, as follow (P22, L12-19):
"OA concentrations in ERSA are also comparable to those observed between June 2012 and July 2013 by Minguillon et al. (2015) at a site in northern Spain 25km from the Mediterranean coast (OA=3.8 µg m$^{-3}$ on average); or to those measured by Debevec et al. (2017) in the eastern basin in Cyprus (OA=3.33 µg m$^{-3}$ on average). Comparable concentrations for ammonium and sulphate were also found by Minguillon et al. (2005) (on average 0.8 and 1.3 µg m$^{-3}$, respectively), while they observed higher nitrate concentrations (0.8 µg m$^{-3}$ on

average). It is worth noting that Minguillon et al. (2005) report yearly measurements and not only summer measurements as in this study."

*Please, could you indicate what is the percentage of the NR-PM1 with respect to the total PM1?*

Based on the correlation obtained between NR PM1 (ACSM) and PM1 (SMPS) (see supplementary information) and the slope close to one between these two datasets, we can conclude that ACSM was in capacity to document the main submicron chemical constituents (no significant influence of sea salt or dust aerosols).

*P26 L4: Solar radiation may also influence biogenic VOCs*

We agree with the second referee and modified the statement in the revised manuscript as follow (P26, L16-17):
"which is well-known to influence biogenic emissions together with solar irradiation (Guenther et al., 1995; 2000)."

*P26 L23-26. As shown in Figure 6 there is not a clear anti-correlation between Factor 6 and wind speed. Even, during specific days and increase in wind speed is related with an increase in Factor 6.*

We agree with second referee that no clear anti-correlation is observed between factor 6 and wind speed. However, it remains true to say that the lowest levels of factor 6 observed on 23, 24 and 25 July coincide with the highest wind speed values, while lower wind speed are observed on average the days when high factor 6 levels are observed (e.g. 26, 27, 28 July and 02 and 03 August). A direct and clear anti-correlation between these two parameters is not expected since biogenic emissions are rather controlled by temperature and solar irradiation. In addition, the oxidation of primary biogenic compounds is also mainly controlled by solar irradiation. However, low dispersion of air masses, in which fresh and intense emissions of primary biogenic compounds occurred, is certainly needed to observed high secondary biogenic compounds, especially first generation compounds as the one observed in Factor 6.

*Please, rewrite P30 L19-L20: Can the diurnal cycles with maxima at midday related to specific transport scenarios (breeze)?*

From our point of view, the phase difference between SV-OOA and LV-OOA diurnal profiles maxima, as shown in the Fig S9, clearly indicates the local photochemical formation of fresh SV-OOA in the morning followed by a rapid oxidation which could explain the enhancement of LV-OAA in the afternoon; therefore excluding sea breeze influence on these midday maxima.

*P31 L25-34. Authors distinguish two periods with impact of processed anthropogenic /continental air masses: 19 - 27 of July and 30/07-03/08. However, as shown in Figure 12 there are marked differences between these two periods. However, the time evolution of both aerosol component s and VOC factors are very similar for the period 17 – 27 of July, including the calm period. The time evolution of ACSM compounds during this period 17 - 27 of July, could be related to the summer recirculation of air masses as shown in previous papers on ACSM data in the area (Minguillon et al Atmos. Chem. Phys., 15, 6379–6391, 2015, Ripoll et al, Atmos. Chem. Phys., 15, 2935–2951, 2015) and described in Pey et al,*

*Atmos. Res., 94, 422–435, 2009, among others. These scenarios characterized by the recirculation of air masses could explain the mix of long, medium and short lived VOCs*

These scenarios have been added to the revised manuscript to explain similarities in aerosol components and VOCs factors during the first part of the campaign (P32, L7-12):
"The evolution of aerosol components and VOCs factors during this period is also similar to the ones observed for the calm low wind conditions at the beginning of the campaign. These similarities could be related to the recirculation of air masses already observed in the western Mediterranean basin causing the formation of reservoir layers at high altitude and described in several studies (Pey et al., 2009; Minguillon et al., 2015; Ripoll et al., 2015)."

*Table1: Increase width of first column*

The width of first column of table 1 has been increased in the revised manuscript.